# Glutamatergic projections from the substantia nigra pars reticulata to the dorsal raphe nucleus regulate male social hierarchies

Yanzhu Fan[1], Shaoxiang Ge[1,2], Lidi Lu[1], Wenjun Niu[1], Zhiyue Wang[1], Xiaoguo Jiao[2], Guangzhan Fang[1]*

1 Chengdu Institute of Biology, Chinese Academy of Sciences, Chengdu, Sichuan, China, 2 School of Life Sciences, Hubei University, Wuhan, Hubei, China

* fanggz@cib.ac.cn

## Abstract

Social hierarchy constitutes a fundamental organizational characteristic among various social species, significantly influencing individual survival, health, and reproductive success within these societies. Neurons in the substantia nigra pars reticulata (SNr) exhibit extensive connectivity with the dorsal raphe nucleus (DRN), a critical structure implicated in social interaction, reward processing, and the establishment of social rank. However, the specific neuronal types within the SNr, as well as the associated neural circuits that regulate social dominance, remain inadequately characterized. This study aims to elucidate the crucial role of SNr glutamatergic (SNr$^{Glu}$) neurons in the establishment and maintenance of social hierarchy in male mice. Employing fiber photometry, we observed that the activation of SNr$^{Glu}$ neurons increased during the initiation of effortful behaviors in the tube test. Further investigations revealed that optogenetic activation or chemogenetic inhibition of the SNr$^{Glu}$ neurons induced upward or downward shifts in social ranks, respectively. Additionally, our findings indicate that the activation of SNr glutamatergic terminals in DRN elevates social status and reduces anxiety levels in mice. Collectively, these results broaden our understanding of the functions associated with SNr$^{Glu}$ neurons and underscore their critical role in regulating social hierarchy among male mice. This work enhances our understanding of the functions of SNr$^{Glu}$ neurons in both physiological contexts and neurological disorders.

## Introduction

The social hierarchy serves as a fundamental organizing principle in many animal societies, including insects, fish, reptiles, rodents, and primates [1–3]. This organizational framework delineates individual rankings within a group, established through recurrent social interactions such as aggressive encounters or behavioral displays that convey information regarding fighting ability or dominance status [4–6]. An

**Data availability statement:** The dataset generated and analyzed during the current study is available in the Dryad repository: https://doi.org/10.5061/dryad.m0cfxppg3. Furthermore, the scripts for data visualization are accessible in the Zenodo repository: https://doi.org/10.5281/zenodo.18618228.

**Funding:** This study was supported by the Natural Science Foundation of Sichuan Province, China (http://202.61.89.120/) (2026NSFSC0920 to YZF), the Major Science and Technology Projects of Henan Province, China (https://kjt.henan.gov.cn/?zbb=true) (221100210500 to GZF), and the National Natural Science Foundation of China (https://www.nsfc.gov.cn/) (32170504 to GZF). The funders had no role in study design, data collection and analysis, decision to publish, or preparation of the manuscript.

**Competing interests:** The authors have declared that no competing interests exist.

Abbreviations: CPu, caudate putamen; DRN, dorsal raphe nucleus; EPM, elevated plus-maze; i.p., intraperitoneal; MDT, mediodorsal thalamus; OFT, open field test; PBS, phosphate buffer solution; PETH, peri-event time histogram; PFC, prefrontal cortex; SNr, substantia nigra pars reticulata; SNr$^{Glu}$, SNr glutamatergic; VPM, ventral posteromedial thalamic nucleus; ZID, zona incerta, dorsal part; 5-HT, 5-HT serotonin.

individual's position within this hierarchy is frequently associated with various fitness-related outcomes. For example, social dominance status significantly influences an individual's health, survival, and reproductive success [7–14]. Moreover, all group members benefit from the stability provided by dominance hierarchies, which mitigate agonistic behaviors and, in turn, conserve energy and reduce the risk of physical injury [7,15–19]. Consequently, elucidating the central neural mechanisms underlying social hierarchies is of paramount importance for comprehending both individual life histories and population dynamics. Despite this significance, the neural mechanisms by which dominance hierarchies are established and maintained remain inadequately understood.

Numerous brain regions have been implicated in the regulation of social dominance, including the prefrontal cortex (PFC), amygdala, hippocampus, hypothalamus, striatum, and lateral habenula [4,19–23]. Notably, the PFC has been identified as the principal regulatory center for social hierarchy, processing information regarding social status received from upstream brain regions and modulating the dominance behaviors through downstream areas [23–26]. Specifically, a recent study has shown that the dmPFC-DRN pathway promotes social competition [27]. Among the identified neural circuits relevant to social dominance, the reward- and motivation-related mesolimbic dopamine system is particularly prominent [28]. In murine models, social submission appears to coincide with a reduction in motivational drive, suggesting that social dominance is intricately linked to motivational processes [29]. Importantly, motivational and reward-related processes are largely regulated by the mesolimbic system, leading to the inference that this system may play a critical role in the formation of social hierarchies. Consistent with this notion, various subcortical structures, including the substantia nigra pars reticulata (SNr), dorsal raphe nucleus (DRN), and amygdala, are involved in the representation of social rank signals in some species [21]. For instance, dopamine signaling is elevated in response to alterations in status signals within the SNr of green anole lizards (*Anolis carolinensis*) [30], indicating a potential role for the SNr in the establishment and maintenance of dominance hierarchies.

The SNr functions as a pivotal hub for the integration of diverse upstream input signals, coordinating multifaceted functions such as motor control, cognition, habit formation, and reward [31–44]. While the majority of neurons in the SNr are GABAergic, the population also includes glutamatergic and dopaminergic neurons, as well as those utilizing both glutamate and dopamine as neurotransmitters. In comparison to the well-characterized GABAergic neurons of the SNr, the role of glutamatergic neurons remains less elucidated; however, some SNr glutamatergic (SNr$^{Glu}$) neurons may share developmental and functional relationships with their GABAergic counterparts. The DRN, a critical component of the mesolimbic system, has been identified as the primary source of serotonin (5-HT) in the brain [45–47]. Previous studies have established a robust association between the serotoninergic system and social rewards, as well as the representation of social status across a variety of species, including crustaceans, reptiles, rodents, and primates [20,48–57]. For instance, the injection of 5-HT into the hemolymph of both lobsters and crayfish reduces the

likelihood of retreat and extends the duration of aggressive interactions [58,59]. Moreover, experimentally enhancing serotonergic activity has been shown to facilitate the acquisition of dominance in monkeys, whereas reductions in this activity correspondingly diminish dominance [53]. Similar effects have been documented in humans, where the administration of 5-HT influences social dominance [60]. Collectively, these findings underscore the potential role of the DRN and the serotonergic system in the regulation of social dominance. Notably, both serotonergic and GABAergic neurons within the DRN receive glutamatergic inputs from the SNr [61,62]. Thus, we hypothesize that the glutamatergic pathway extending from the SNr to the DRN (SNr$^{Glu}$-DRN) may play a significant role in regulating social hierarchy.

To investigate this hypothesis, we employed fiber photometry to measure the activity patterns of SNr$^{Glu}$ neurons in conjunction with specific behaviors exhibited by male mice in the competitive tube test. The results indicated heightened activity levels of these neurons during instances of effortful behavior in the tube test. Furthermore, we utilized cell-type-specific optogenetic and chemogenetic methodologies to manipulate SNr$^{Glu}$ neurons and evaluate their role in social competition. In addition, our findings demonstrate that optogenetic activation of the SNr$^{Glu}$-DRN pathway induces an increase in social ranks. Collectively, these findings expand our understanding of the functions associated with the glutamatergic pathway from the SNr to the DRN, highlighting its potential role in the regulation of social hierarchy among male mice.

## Results

### Population activity of SNr$^{Glu}$ neurons increased during effortful behavioral epochs in social contest scenarios

To investigate the real-time population activity of SNr$^{Glu}$ neurons during social competition in the tube test, we bilaterally administered rAAV-CaMKIIα-GCaMP6f, which encodes the fluorescent calcium indicator GCaMP6f, or rAAV-CaMKIIα-EYFP into the SNr of mice (Fig 1A). Histological analyses demonstrated that GCaMP6f or EYFP localization was predominantly restricted to the SNr (S1A and S1B Fig). Additionally, co-labeling with CaMKIIα (Figs 1B, S2A and S2B) or VGLUT2 antibodies (S3A and S3B Fig) confirmed that the majority of these neurons were glutamatergic. Subsequently, we bilaterally implanted optical fibers to facilitate the recording of calcium signals from SNr$^{Glu}$ neurons, while concurrently utilizing a camera to monitor the animals' behavior (Fig 1C). By capturing calcium signals from a randomly selected unilateral SNr, we observed a marked increase in SNr$^{Glu}$ neuronal activity immediately upon the initiation of pushing and counter-pushing during direct confrontations with an opponent in the tube (Fig 1D). In contrast, no significant alterations in calcium signals were detected when mice traversed the tube without any opponent or when EYFP-expressing mice were subjected to the tube test (Fig 1E and 1F). The observed Ca$^{2+}$ dynamics implying that SNr$^{Glu}$ neurons are dynamically activated during effortful behavioral epochs, such as push-initiation and push-back, and that their population activity is correlated with active social contests. To investigate whether the population activity of SNr$^{Glu}$ neurons is specifically linked to physical efforts in a social context related to the representation of social rank, we conducted both an active physical effort task devoid of social components (the push ball test) and a social, noncompetitive effort task (the social interaction test). No significant changes in calcium signals associated with pushes were observed in GCaMP6f-expressing or EYFP-expressing mice during the push ball test ($p > 0.05$; permutation test; S4 Fig). Likewise, no significant alterations in calcium signals related to effortful social interactions were detected in either group of mice throughout the social interaction test ($p > 0.05$; permutation test; S5 Fig). These results suggest that calcium activity levels in SNr$^{Glu}$ neurons are not correlated with physical effort in the absence of social competition. Collectively, these findings underscore the relationship between SNr$^{Glu}$ neuronal activity and social competition and status.

### Optogenetic activation of SNr$^{Glu}$ neurons sufficiently enhances social status in the tube test

To investigate the behavioral responses elicited by the activation of SNr$^{Glu}$ neurons, we bilaterally delivered rAAV-CaMKIIα-hChR2(H134R)-EYFP or rAAV-CaMKIIα-EYFP constructs to the SNr of mice (hereafter referred to as ChR2-EYFP mice and EYFP mice, respectively; see Fig 2A). Histological analyses revealed that the localization of ChR2-EYFP was primarily confined to the SNr (S1C Fig). Furthermore, colocalization with CaMKIIα (Figs 2B and S2C) or VGLUT2 antibodies (S3C Fig)

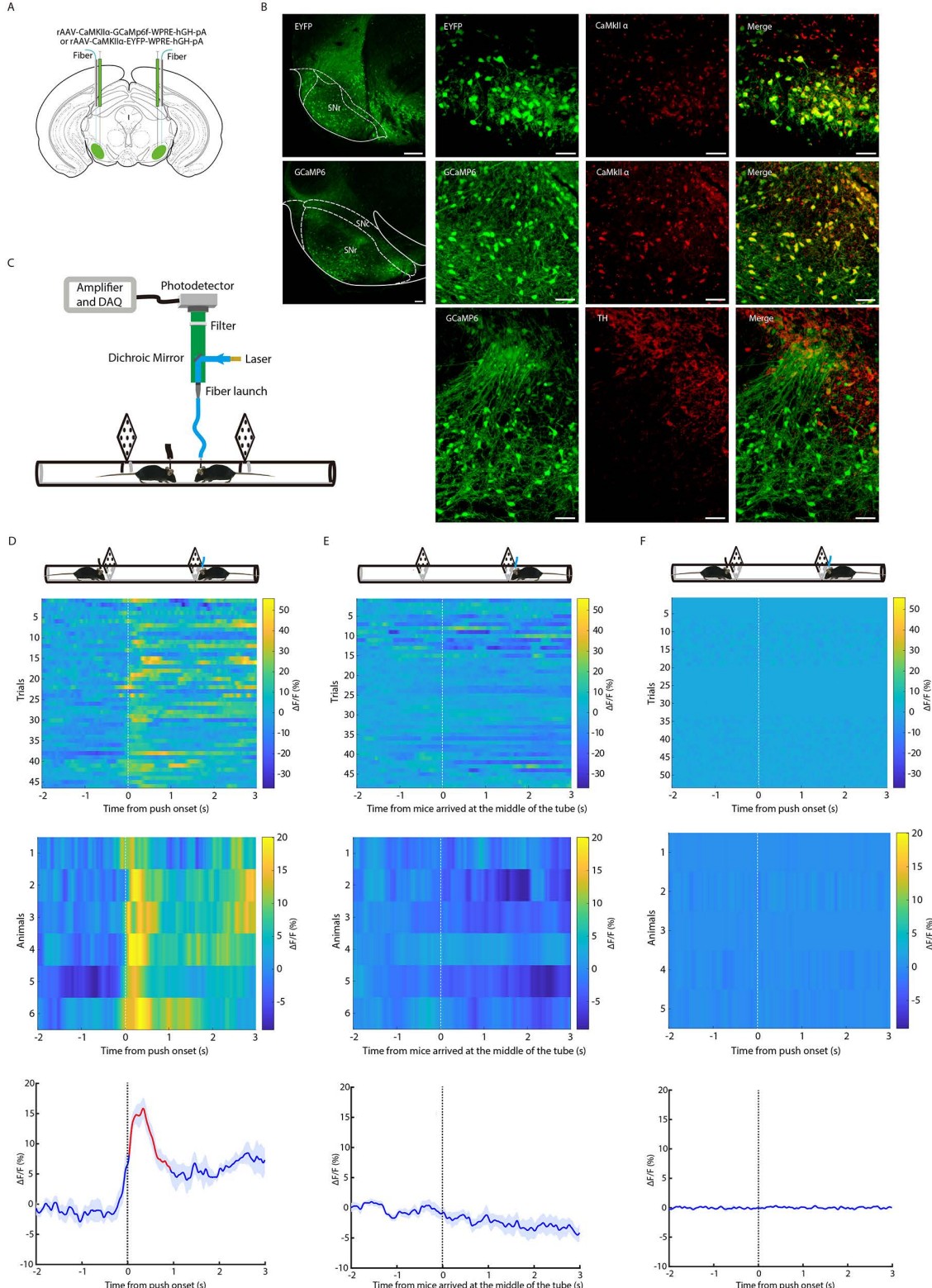

**Fig 1. Ca²⁺ activity of SNrᴳˡᵘ neurons during push-initiation and push-back in the tube test. (A)** Schematic representation of the CaMKIIα-GCaMP6f viral construct, injection site, and optic fiber placement over the SNr. **(B)** Left, a representative coronal section of the SNr shows the

expression of EYFP or GCaMP6f. Scale bar, 200 μm. Right, immunostaining confirming the specificity of EYFP and GCaMp6f expression in glutamatergic neurons (not dopaminergic neurons). Scale bar, 60 μm. **(C)** Schematic of the fiber photometry setup used to record SNr$^{Glu}$ neuron calcium activity during the tube test. **(D)** Ca$^{2+}$ signals aligned with the initiation of pushes when mice expressing GCaMP6f encounter their cagemates. The upper panel shows a heatmap of Ca$^{2+}$ signals aligned to push-initiation and push-back, each row represents one trial ($n = 46$ trials from 6 mice). The middle panel displays a heatmap representing the average Ca$^{2+}$ signals for each animal across all trials ($n = 6$). The lower panel shows mean Ca$^{2+}$ transients associated with pushes for the entire test group ($n = 6$), with solid line indicating mean and shaded areas indicating SEM. A red segment denotes a statistically significant increase from baseline ($p < 0.05$; permutation test). **(E)** Ca$^{2+}$ signals aligned with the time point when mice reached the middle of the tube without an opponent. The upper panel presents data from all trials ($n = 48$), the middle panel from all animals ($n = 6$), and the lower panel shows the mean ± SEM of the average Ca$^{2+}$ signals. **(F)** Ca$^{2+}$ signals aligned with the initiation of pushes in mice expressing EYFP encountering their cagemates. The upper panel presents data from all trials ($n = 53$), the middle panel from all animals ($n = 5$), and the lower panel shows the mean ± SEM of the average Ca$^{2+}$ signals. Abbreviations: SNc, substantia nigra pars compacta; SNr, substantia nigra pars reticulata; TH, tyrosine hydroxylase. The data underlying this Figure can be found in files numbered 1 to 3 on Dryad (https://doi.org/10.5061/dryad.m0cfxppg3).

confirmed that the majority of these neurons were glutamatergic. Subsequently, we implanted optic fibers above the injection sites. Following a minimum of 4 weeks post-viral injection, we stimulated the randomly selected unilateral SNr in vivo using pulses of 473 nm blue light (20 ms duration, 20 Hz frequency, and 10–20 mW intensity) prior to the mice entering the tube. Photostimulation of the unilateral SNr significantly influenced behavioral performance in the tube test (Fig 2D-2H), resulting in elevated tube test rankings for ChR2-EYFP mice (Fig 2I-2K), whereas no such effects were observed in EYFP mice (S6 Fig). In ChR2-EYFP mice, photostimulation markedly enhanced effortful behaviors, including push-initiation (Fig 2D), push-back (Fig 2E), and duration of stillness (Fig 2G), while concurrently reducing the incidence of retreats (Fig 2F). Notably, photostimulation did not affect the number of stillness (Fig 2G) or the proportion of time spent resisting (Fig 2H). Compared to controls, the significantly elevated social ranks were maintained for at least 4 days (Fig 2I-2K). Collectively, these findings suggest that optogenetic activation of SNr$^{Glu}$ neurons in mice promotes an increase in social hierarchy.

Subsequently, we conducted a series of behavioral assays to further explore the association between various behavioral performances and the optogenetic activation of SNr$^{Glu}$ neurons in mice. During light-on periods, the locomotion of mice in the open field test (OFT) significantly increased (Fig 2L). In contrast, the number of entries into the central area and the time spent in the central area did not exhibit a significant change compared to light-off conditions. These findings suggest that the activation of SNr$^{Glu}$ neurons is sufficient to enhance locomotor behavior. Notably, the duration of time spent in the open arms prior to light activation in the elevated plus-maze (EPM) test was significantly greater than that observed during and after light-on periods; however, no differences in open arm time were found between ChR2-EYFP and EYFP mice (S7A Fig), indicating that optogenetic activation of SNr$^{Glu}$ neurons, rather than light exposure alone, does not influence anxiety-like behaviors. Furthermore, SNr-photostimulated mice exhibited a typical preference for novel mice in the social memory test (S7B Fig) and displayed normal levels of both aggressive and nonaggressive behaviors toward novel mice in the resident-intruder test (S7C Fig). Additionally, activation of SNr$^{Glu}$ neurons did not impact muscle strength (S7D Fig). These results collectively indicate that the alterations in social hierarchy observed in mice following the optogenetic activation of SNr$^{Glu}$ neurons are not attributable to changes in social preference, aggression levels, or forelimb grip strength. Importantly, the push ball test demonstrated that optogenetic activation of SNr$^{Glu}$ neurons did not significantly affect either the latency required to move the ball out of the tube or the number of active pushes exhibited by the mice (S7E Fig). Similarly, the social interaction test indicated that the same activation did not influence the mice's effort-based social behavior or their nonsocial exploratory activities (S7F Fig). Collectively, these findings suggest that optogenetic activation of SNr$^{Glu}$ neurons does not influence physical effort in the active task devoid of social components, nor does it affect physical effort in the social yet noncompetitive context.

## Chemogenetic inhibition of SNr$^{Glu}$ neurons results in defeat in the tube test

To further validate the critical role of SNr$^{Glu}$ neurons within the competitive context of the tube test, we employed the inhibitory DREADD (designer receptors exclusively activated by designer drugs) system. Specifically, we bilaterally

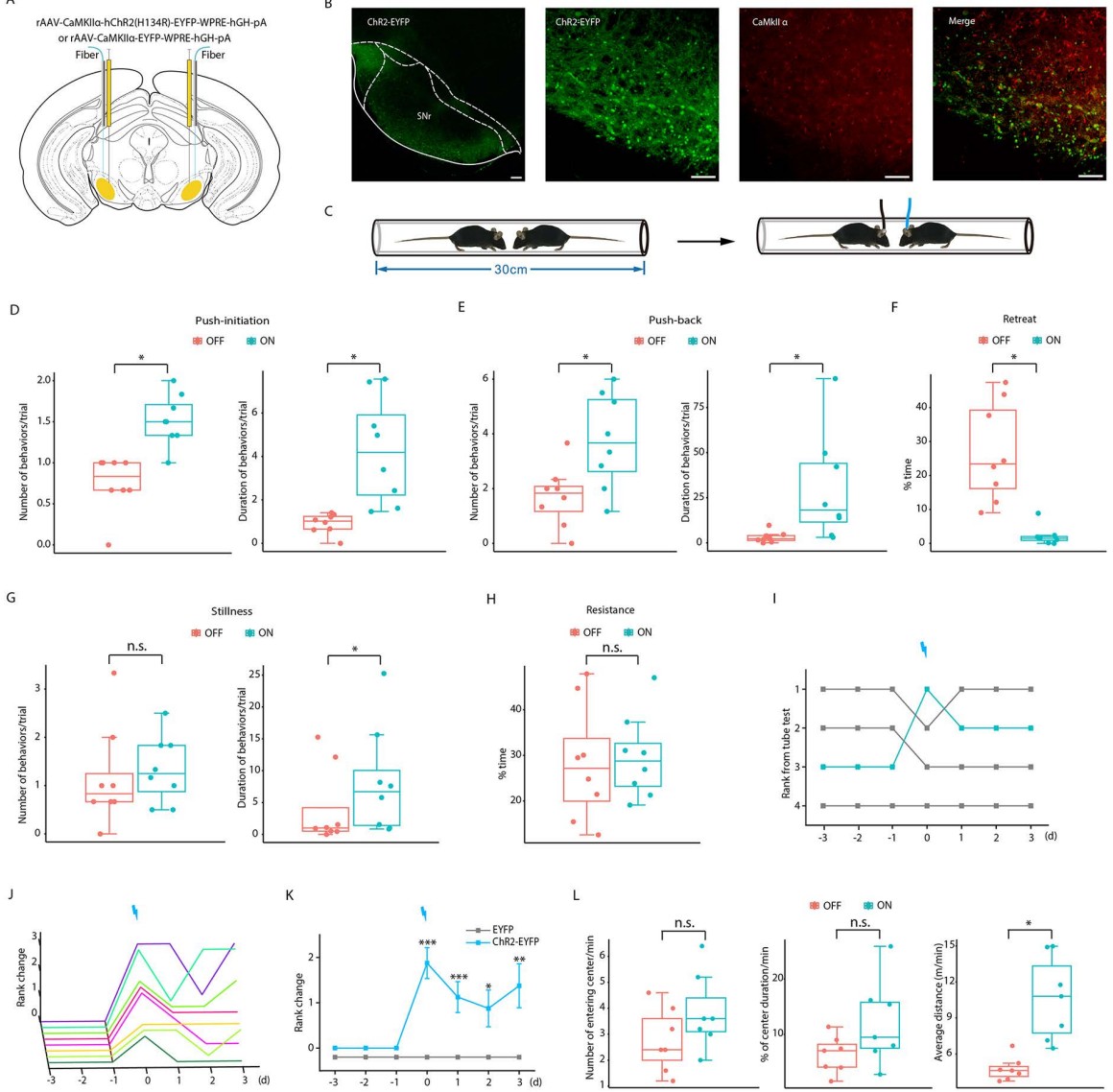

**Fig 2. Optogenetic activation of SNr<sup>Glu</sup> neurons induces winning in the tube test. (A)** A schematic representation illustrating the ChR2-EYFP and EYFP viral constructs, injection sites, and optic fiber positioning above the SNr. **(B)** Left, a representative coronal section of the SNr shows the expression of ChR2-EYFP. Scale bar, 200 μm. Right, immunostaining confirming the specificity of ChR2-EYFP expression in glutamatergic neurons. Scale bar, 60 μm. **(C)** Diagrams illustrating the tube test with and without optogenetic activation. **(D)** Quantification of push-initiation occurrences and durations for each animal in the tube test ($n = 8$, Wilcoxon signed rank test; number: $Z = -2.533$, $p = 0.011$; duration: $Z = -2.521$, $p = 0.012$). **(E)** Analysis of push-back occurrences and durations for each animal in the tube test ($n = 8$, Wilcoxon signed rank test; number: $Z = -2.380$, $p = 0.017$; duration: $Z = -2.521$, $p = 0.012$). **(F)** Percentage of time spent retreating ($n = 8$, Wilcoxon signed rank test; $Z = -2.521$, $p = 0.012$). **(G)** Quantification of stillness occurrences and durations for each animal in the tube test ($n = 8$, Wilcoxon signed rank test; number: $Z = -0.844$, $p = 0.398$; duration: $Z = -2.240$, $p = 0.025$). **(H)** Percentage of time spent resisting ($n = 8$, Wilcoxon signed rank test; $Z = -0.280$, $p = 0.779$). **(I)** An example demonstrating how the third-ranked mouse in a given cage achieved first place following photostimulation, assessed daily over 7 days. **(J)** Summary of rank changes in ChR2-EYFP mice pre- and post-optogenetic activation of SNr<sup>Glu</sup> neurons, with each line representing an individual animal. **(K)** Difference in average rank change between ChR2-EYFP and EYFP mice after optogenetic activation of SNr<sup>Glu</sup> neurons ($n = 8$ for ChR2-EYFP mice and $n = 9$ for EYFP mice, Mann–Whitney $U$ test; Day 0: $U = 0$, $p < 0.001$; Day 1: $U = 4.5$, $p < 0.001$; Day 2: $U = 13.5$, $p = 0.007$; Day 3: $U = 9$, $p = 0.002$). **(L)** Optogenetic activation of SNr<sup>Glu</sup> neurons increased locomotion in the open field test (average distance: $n = 7$, Wilcoxon signed rank test, $Z_6 = -2.366$, $p = 0.018$), but did not affect the number of entries into the central area ($n = 7$, paired two-sided $t$ test, $t_6 = -1.887$, $p = 0.108$) and center time ($n = 7$, Wilcoxon signed rank test, $Z = -1.183$, $p = 0.237$). * $p < 0.05$; ** $p < 0.01$; *** $p < 0.001$. The data underlying this Figure can be found in files numbered 4 to 8 on Dryad (https://doi.org/10.5061/dryad.m0cfxppg3).

administered rAAV-CaMKIIα-hM4D(Gi)-EGFP or rAAV-CaMKIIα-EYFP into the SNr of mice, subsequently designated as hM4Di-EGFP mice and EYFP mice, respectively (Fig 3A). Histological analyses indicated that hM4Di-EGFP localization was predominantly restricted to the SNr (S1D Fig). Moreover, co-labeling with CaMKIIα (Figs 3B and S2D) and VGLUT2 antibodies (S3D Fig) confirmed that the majority of these neurons were glutamatergic. Following this intervention, Clozapine-N-oxide (CNO, 5 mg/kg) was administered intraperitoneally to each mouse in the experimental group, while their cage mates received an equivalent volume of saline. The tube test was conducted at intervals of 1–1.5, 3–5, 6–8, 24, 48, and 72 hours post-injection. The inhibition of SNr^Glu neurons resulted in significant behavioral alterations and a reduction in tube test rankings among the hM4Di-EGFP mice treated with CNO, in comparison to the EYFP mice receiving CNO (Fig 3C-3K) and the hM4Di-EGFP mice receiving saline (S8 Fig). At 3.5 hours post-injection, both the number and duration of push initiations (Fig 3C) and the duration of immobility (Fig 3E) were significantly diminished in the hM4Di-EGFP mice, while the frequency of push-backs (Fig 3D), resistance (Fig 3F), and retreats (Fig 3G) increased. The significantly reduced rankings compared to controls were sustained for ~8 hours; however, by 24 hours post-CNO administration, most mice reverted to their original rank positions (Fig 3I-3K).

To explore whether the alterations in tube test rankings induced by chemogenetic inhibition of SNr^Glu neurons can be generalized to rankings assessed through other paradigms, we conducted the warm spot test (Fig 3O). Our findings demonstrate that inhibition of SNr^Glu neurons led to a significant reduction in the duration of time spent in the warm spot by hM4Di-EGFP mice at 2 hours post-CNO injection, compared to saline controls (Fig 3P-3Q). In contrast, EYFP mice exhibited no significant changes in warm spot occupancy or social hierarchy following either CNO or saline injections (S9 Fig). These results indicate that social hierarchical status can be generalized across various social contexts in mice. Taken together, our data suggest that chemogenetic inhibition of SNr^Glu neurons contributes to a decline in social dominance, thereby underscoring the essential role of these neurons in the maintenance of social hierarchy.

To assess the potential alterations in exploratory and locomotion behaviors subsequent to the chemogenetic inhibition of SNr^Glu neurons, we conducted an open field test on mice. Our results indicate a significant reduction in the number of entries into the center, time spent in the center, and overall locomotor activity among hM4Di-EGFP mice 2 hours post-CNO injection, in contrast to the saline injection (Fig 3L-3N). Notably, no significant differences in these locomotor parameters were observed in EYFP mice following either CNO or saline injections (Fig 3L-3N). These findings underscore the critical role of SNr^Glu neuron activation in sustaining heightened exploratory and locomotor behaviors. To minimize the potential influence of motor incoordination induced by DREADD manipulation on mouse performance in the tube test, we conducted additional assessments using the pole test and rotarod test 2 hours post-injection of either CNO or saline. The pole test indicated no significant differences in the time taken for mice to transition from the head-up to head-down position, nor in the total duration from head-up to descent at the bottom of the pole, between CNO and saline injections or between hM4Di-EGFP and EYFP mice (S10A and S10B Fig). Similarly, the rotarod test revealed no significant differences in latency, speed, or total distance traveled between the CNO and saline injections, as well as between the hM4Di-EGFP and EYFP mice (S10C–S10E Fig). Collectively, these results suggest that motor coordination in mice was not affected by DREADD-mediated inhibition of SNr^Glu neurons.

**SNr^Glu neurons modulate social hierarchy via the SNr^Glu-DRN pathway**

Next, we investigated the specific output pathways of the SNr, focusing on the glutamatergic neurons. By injecting rAAV-CaMKIIα-EYFP virus into the SNr of wild-type mice, we selectively labeled SNr glutamatergic neurons and their axons with EYFP (Fig 4A). EYFP-expressing SNr glutamatergic axons were observed in the forebrain and brainstem regions, including the caudate putamen (CPu), ventral posteromedial thalamic nucleus (VPM), zona incerta, dorsal part (ZID), and DRN (Fig 4B). To assess whether the specific stimulation of SNr glutamatergic projections can modulate social hierarchy, we stimulated the terminals of SNr^Glu neurons in these regions in ChR2-EYFP mice, using the protocol from the tube test. Optogenetic activation of SNr^Glu-CPu, -VPM, and -ZID pathways did not affect the social hierarchy of mice (S11 Fig).

PLOS Biology

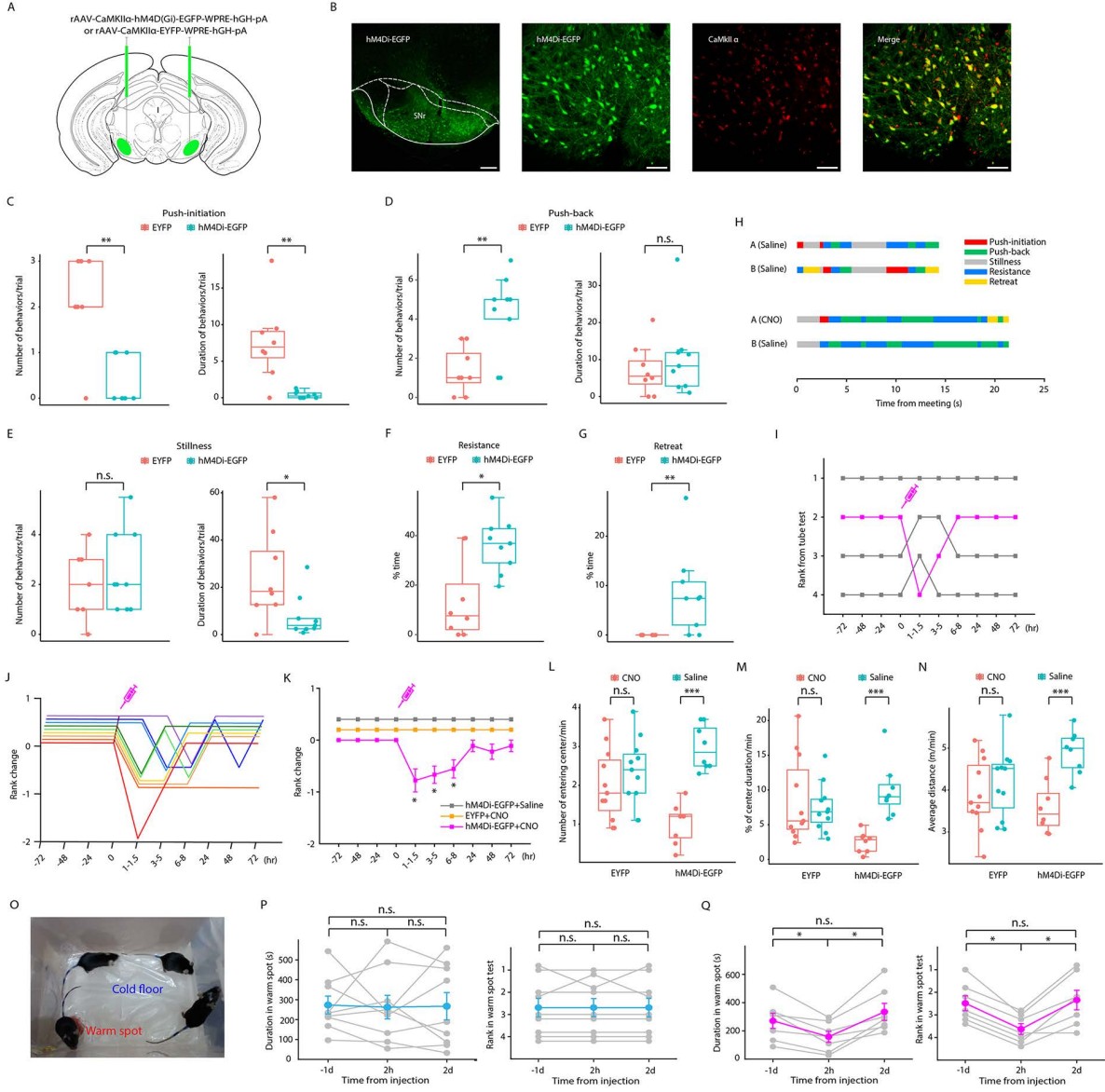

**Fig 3. Chemogenetic inhibition of SNr^Glu neurons results in defeat in the tube test.** **(A)** A schematic representation of the hM4Di-EGFP and EYFP viral construct and their injection sites in the SNr. **(B)** Left, a representative coronal section of the SNr, illustrating hM4Di-EGFP expression. Scale bar, 200 μm. Right, immunostaining confirming the specificity of hM4Di-EGFP expression in glutamatergic neurons. Scale bar, 60 μm. **(C)** Number and duration of push-initiation for each animal in the tube test ($n = 8$ mice for EYFP and $n = 9$ for hM4Di-EGFP, Mann–Whitney $U$ test; number: U=7, $p = 0.004$; duration: U=7, $p = 0.005$). **(D)** Number and duration of push-back for each animal in the tube test ($n = 8$ vs. 9, Mann–Whitney $U$ test; number: U=9, $p = 0.008$; duration: U=30, $p = 0.563$). **(E)** Number and duration of stillness for each animal in the tube test ($n = 8$ vs. 9, Mann–Whitney $U$ test; number: U=31, $p = 0.621$; duration: U=15, $p = 0.043$). **(F)** Percentage of time spent resisting ($n = 8$ vs. 9, Mann–Whitney $U$ test; U=11, $p = 0.016$). **(G)** Percentage of time spent retreating ($n = 8$ vs. 9, Mann–Whitney $U$ test; U=8, $p = 0.003$). **(H)** Behavioral annotation of two tube test trials between the same pair of hM4Di-EGFP mice, pre- and post-CNO injection, with a 1-week interval between trials. **(I)** Tube test outcomes for a cohort of hM4Di-EGFP mice before and after CNO administration to the rank-2 mouse at time 0. **(J)** Summary of rank changes in hM4Di-EGFP mice following CNO administration to each individual at time 0, with each line representing an individual mouse. **(K)** Mean change in rank prior to and following injection of CNO or saline ($n = 9$ for hM4Di+CNO, $n = 7$ for hM4Di+Saline while $n = 8$ for EYFP+CNO, Wilcoxon signed rank test, at 1–1.5 hr: Z=−2.333, $p = 0.020$; at 3−5 hr: Z=−2.449, $p = 0.014$; at 6−8 hr: Z=−2.236, $p = 0.025$). **(L)–(N)** Chemogenetic inhibition of SNr^Glu neurons led to a reduction in entries into the center, time spent in the center, and locomotion in the open field test for hM4Di-EGFP mice (one-way repeated measures ANOVA with Bonferroni correction; number of entries: $n = 8$, $F_{1,7} = 171.726$, $p < 0.001$; center time: $n = 8$, $F_{1,7} = 61.080$, $p < 0.001$; average distance: $n = 8$, $F_{1,7} = 43.053$, $p < 0.001$) but not for EYFP mice ($n = 11$, $p > 0.05$). **(O)** Depiction of the warm spot test, where four mice from the same cage compete for a warm corner in a cage with an ice-cold floor.

(**P**) and (**Q**) Duration (left) and rank (right) in the warm spot test for hM4Di-EGFP mice at 1 day pre-injection, 2 hours post-injection, and 2 days post-injection of saline (duration: $n = 9$, one-way repeated measures ANOVA with Bonferroni correction, $F_{2,16} = 0.023$; $p = 0.978$; rank: Friedman test, $t_2 = 0.667$; $p = 0.717$) or CNO (duration: $n = 7$, $F_{2,12} = 11.902$; $p = 0.001$; rank: $t_2 = 11.043$; $p = 0.004$). * $p < 0.05$; ** $p < 0.01$; *** $p < 0.001$. The data underlying this Figure can be found in files numbered 9–14 on Dryad (https://doi.org/10.5061/dryad.m0cfxppg3).

However, activation of the SNr^Glu-DRN pathway increased effortful behaviors and enhanced social ranking in ChR2-EYFP mice (Fig 4C-4H), consistent with the effects observed upon photostimulation of SNr^Glu neurons in the SNr, while no similar effects were observed in EYFP control mice (S12 Fig). In ChR2-EYFP mice, photostimulation of SNr^Glu terminals in the DRN significantly increased effortful behaviors, including push-initiation (Fig 4C) and push-back (Fig 4D), while concurrently decreasing the frequency of retreats (Fig 4E). However, this photostimulation did not significantly impact stillness (S13A Fig) or resistance (S13B Fig). Notably, the observed elevation in social status was sustained for at least 3 days in comparison to the control group (Fig 4G-4H). Collectively, these findings indicate that optogenetic activation of the SNr^Glu-DRN pathway is sufficient to enhance social dominance.

To elucidate the behavioral implications associated with optogenetic activation of the SNr^Glu-DRN pathway in mice, we next conducted a series of behavioral assays. Notably, we observed that the duration spent in the open arm prior to light activation during the EPM test was significantly greater than that observed during and after light activation (Fig 4I). In comparison to EYFP mice, photostimulation of the terminals of SNr^Glu neurons in the DRN resulted in increased duration in the open arm for ChR2-EYFP mice (Fig 4I), suggesting that optogenetic activation of the SNr^Glu-DRN pathway may modulate anxiety levels in mice. Furthermore, photostimulation of this pathway preserved typical preferences for novelty in the social memory test (Fig 4J), and exhibited a standard range of aggressive and nonaggressive behaviors towards novel conspecifics in the resident-intruder test (S13C Fig). Additionally, metrics such as the number of entries into the central area, time spent in the central area, and locomotion in the OFT under light exposure did not differ significantly compared to the light-off condition (S13D Fig). Photostimulation of the SNr^Glu-DRN pathway did not influence forelimb muscle strength (S13E Fig). Importantly, the push ball test demonstrated that optogenetic activation of SNr^Glu-DRN pathway did not significantly alter either the latency required to move the ball out of the tube or the number of active pushes exhibited by the mice (S13F Fig). Similarly, the social interaction test revealed that the same activation did not affect the mice's effort-based social behaviors or their nonsocial exploratory activities (S13G Fig). Consequently, these findings suggest that optogenetic activation of SNr^Glu-DRN pathway does not impact physical effort in the task that lack social components, nor does it influence physical effort in social contexts that are noncompetitive. Collectively, the current findings suggest that the SNr^Glu-DRN circuit may play a potential role in the regulation of both social hierarchy and anxiety levels, thereby indicating a potential link between alterations in social hierarchy and anxiety in mice.

## Discussion

As the largest output nucleus of the basal ganglia, the SNr integrates a multitude of upstream input signals and projects to various brainstem regions, functioning as a central hub for the signal integration and coordination of these targets' functions [31,32]. Anatomically, the SNr is organized into parallel modules that influence both ascending and descending targets implicated in the regulation of behaviors such as motor control, cognition, habit formation, and reward processing [33–44]. Crucially, the SNr occupies a strategic position in the processing of reward-related information, which guides motor responses to secure future rewards [41,63]. Given that social interactions related to dominance hierarchies represent one aspect of a broader spectrum of reward-related behaviors, it is plausible that the activities of SNr neurons are associated with social interactions. Consistent with this notion, our findings reveal that the activation and inhibition of SNr^Glu neurons exert opposing effects on competitive outcomes. Notably, the heightened activity of SNr^Glu neurons during effortful behaviors underscore their critical role in mediating social hierarchy, further reinforcing the essential functions of the SNr in various physiological regulation processes [31,32]. It is noteworthy that calcium activity levels in SNr^Glu neurons

PLOS Biology | https://doi.org/10.1371/journal.pbio.3003687   March 3, 2026
9 / 26

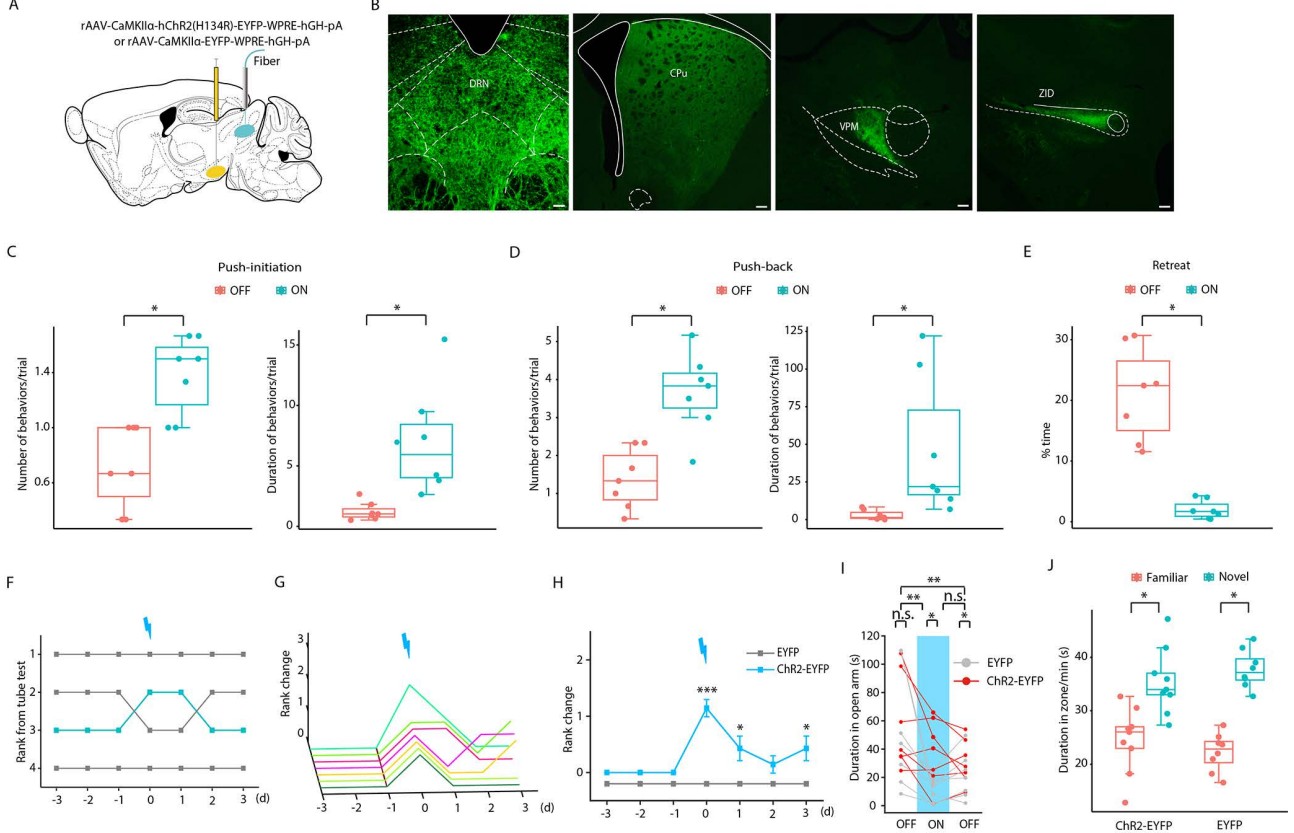

**Fig 4. Optogenetic activation of SNr$^{Glu}$-DRN pathway induces winning in the tube test. (A)** A schematic representation of ChR2-EYFP and EYFP viral constructs, detailing their injection sites within the SNr, alongside the optic fiber placement above the DRN. **(B)** Representative SNr terminal sites expressing EYFP. Scale bar, 200 μm. **(C)** Number and duration of push-initiation for each animal in the tube test ($n=7$, Wilcoxon signed rank test; number: Z=−2.375, $p=0.018$; duration: Z=−2.366, $p=0.018$). **(D)** Number and duration of push-back for each animal in the tube test ($n=7$, Wilcoxon signed rank test; number: Z=−2.366, $p=0.018$; duration: Z=−2.366, $p=0.018$). **(E)** Percentage of time spent retreating ($n=7$, Wilcoxon signed rank test; Z=−2.366, $p=0.018$). **(F)** Example of rank position dynamics over a 7-day period within a cohort of mice, illustrating the ascension of the third-ranked mouse to second position following photostimulation of SNr$^{Glu}$ neuron terminals in the DRN. **(G)** Summary of rank changes in ChR2-EYFP mice pre- and post-optogenetic activation of the SNr$^{Glu}$-DRN pathway. Each trajectory represents a single animal. **(H)** Comparison of average rank changes between ChR2-EYFP and EYFP mice post-optogenetic activation of the SNr$^{Glu}$-DRN pathway ($n=7$ for the ChR2-EYFP mice and $n=8$ for the EYFP mice, Mann–Whitney $U$ test; Day 0: U=0, $p<0.001$; Day 1: U=16, $p=0.046$; Day 2: U=24, $p=0.285$; Day 3: U=16, $p=0.046$). **(I)** In the elevated plus-maze test, the time spent in the open arms decreased over time ($n=7$ for the ChR2-EYFP mice and 7 for EYFP mice, two-way repeated measures ANOVA with Bonferroni correction; $F_{2,24}=15.257$, $p<0.001$), with ChR2-EYFP mice spending more time in the open arms compared to EYFP controls ($F_{1,12}=6.610$, $p=0.024$). **(J)** Both ChR2-EYFP and EYFP mice demonstrated a significant preference for novel mice during the social memory test under light-on conditions ($n=9$ for ChR2-EYFP mice and 8 for EYFP mice, two-way repeated measures ANOVA with Bonferroni correction; $F_{1,15}=28.896$, $p<0.001$), with no significant difference observed between the groups ($F_{1,15}=0.052$, $p=0.822$). Abbreviations: CPu, caudate putamen (striatum); VPM, ventral posteromedial thalamic nucleus; ZID, zona incerta, dorsal part. *$p<0.05$; **$p<0.01$; ***$p<0.001$. The data underlying this Figure can be found in files numbered 15–18 on Dryad (https://doi.org/10.5061/dryad.m0cfxppg3).

are not correlated with physical effort in the active task devoid of social components, nor with physical effort in the social yet noncompetitive context. Furthermore, optogenetic activation of SNr$^{Glu}$ neurons did not influence effort-related behaviors that are independent of social competition. These findings further corroborate the role of SNr$^{Glu}$ neurons in regulating social hierarchy among mice. Consequently, our study elucidates a causal relationship between social dominance and SNr$^{Glu}$ neurons, highlighting the fundamental role of the SNr in both social dominance and effortful behaviors associated with social competition. Nevertheless, the quantitative relationship among the activity of SNr$^{Glu}$ neurons, the social

hierarchies of mice, and their physical exertion requires further exploration. Investigating this interplay may provide deeper insights into the mechanisms by which SNr[Glu] neurons regulate the formation and maintenance of social hierarchies in mice.

The current results indicate that activation of the SNr[Glu]-DRN pathway enhances hierarchical dominance in mice, consistent with previous studies highlighting the role of the DRN in social interactions, social competition, reward processing, cognition, and aggressive behaviors [27,64,65]. The serotoninergic system is closed associated with social rewards and the representation of social status across various species, including crustaceans, reptiles, rodents, and primates [20,48–57]. For instance, the injection of 5-HT into the hemolymph of lobsters and crayfish reduces the likelihood of retreat and prolongs aggressive encounters [58,59]. Similarly, dominant male vervet monkeys (Cercopithecus aethiops) exhibit approximately double the concentration of 5-HT compared to subordinate individuals [66]. Furthermore, experimentally enhancing serotonergic activity facilitates the acquisition of dominance in monkeys, while reductions in this activity diminish dominance [53]. Comparable effects have been documented in humans, where administration of 5-HT similarly impacts social dominance [60]. Collectively, these studies underscore the significance of the serotonergic system in regulating social dominance and status. The DRN serves as the primary source of 5-HT in the brain, with serotonergic neurons constituting two-thirds of the total neuronal population and establishing extensive forebrain connections [45–47]. Recent investigations have identified that the SNr provides glutamatergic input to both serotonergic and GABAergic neurons within the DRN [61,62]. In this context, our study demonstrates that the SNr[Glu]-DRN pathway may play a role in regulating hierarchy dominance in male mice. However, further research is necessary to elucidate the specific types of downstream neurons and the synaptic connections implicated in this function.

Consistent with previous findings [25,67,68], the present results demonstrate a lasting impact of optogenetic manipulation on alterations in social hierarchy, particularly when compared to chemogenetic activation. The establishment of a dominance hierarchy can be attributed to a reinforcing mechanism known as the "winner effect", in which prior victories increase the likelihood of future successes, thereby significantly influencing the dynamics of social hierarchy and competitive motivation. In mice, this winner effect can be mediated by synaptic strengthening within the circuit connecting the mediodorsal thalamus (MDT) to dmPFC [25]. Due to its millisecond-scale precision in neuronal activation, optogenetics likely facilitates the winner effect, which is supported by synaptic strengthening within the relevant neural circuits [25,68]. Consequently, the enduring impact of optogenetics on social rank arises not from a prolonged duration of neuronal activity, but rather from its precise timing during critical behavioral events, enhancing synaptic plasticity within the associated neural circuitry. In contrast, although chemogenetic activation may have a longer-lasting effect, it lacks the temporal specificity necessary for inducing synaptic plasticity. Further research is warranted to investigate whether the mechanisms of synaptic plasticity within the circuit associated with the SNr also contribute to the winner effect.

Hierarchical position significantly influences physiological responses, behavior, and mental health, thereby affecting an individual's overall emotional state [69,70]. Specifically, social hierarchy plays a crucial role in determining susceptibility to stress and is recognized as a critical risk factor for the development of psychiatric disorders [54,71]. In numerous species, a U-shaped relationship exists between stress hormone levels and social rank [21]. While dominant individuals may enjoy privileged access to resources, they also bear the burden of maintaining their status, which can lead to increased vigilance and stress [72,73]. Conversely, subordinate individuals often face challenges associated with restricted access to resources and limited social interactions, potentially inducing anxiety due to uncertainty or perceived threats [74,75]. This dynamic could lead to an elevated risk of chronic social defeat stress [71] and stress-related psychiatric conditions such as depression and anxiety [72]. Social dominance can be attained through competitive interactions, yet both stress and anxiety can influence the formation of social hierarchies, underscoring their critical roles in determining competitive outcomes among animals [7]. Recent studies in both humans [76] and rodents [77,78] have identified trait anxiety as a crucial predictor of social competitiveness. In humans, individuals of lower social status experience a disproportionately high prevalence of mental disorder [79,80], and exhibit increased mortality rates compared to their higher-ranking counterparts [7,28]. In rat models, highly

anxious individuals typically lose territorial contests against low-anxious conspecifics [75], while low-anxious individuals are more likely to achieve dominance when cohabitating for extended periods [81]. Consistent with these observations, the current findings indicate that the activation of glutamatergic projections from the SNr to the DRN not only enhances social status in mice but also alleviates anxiety levels. This suggests a mutually reinforcing relationship between low-anxious status and the acquisition of dominance. Collectively, these findings highlight the SNr[Glu]-DRN pathway as a potential neural circuit for modulating anxiety and social dominance in male mice, thereby providing insight into the intricate relationship between social hierarchy and anxiety levels [13]. However, we are unable to fully disentangle the causal relationships among neural pathways, social hierarchy, and anxiety levels. This challenge stems from the difficulties involved in transitioning from the activation of a specific set of neurons to the understanding of an ethological phenotype [82]. Contributing factors include the distinction between causal-mechanistic explanations and the neural mechanisms that underpin behavior [82], the tendency of experimental designs to oversimplify behavioral complexity [83], and a granularity mismatch between neural and behavioral data [84]. Furthermore, the interwoven relationship between anxiety and social hierarchy observed in our results further complicates interpretation. Considering that anxiety may regulate individuals' competitiveness [76], we cannot discount the possibility that alterations in anxiety levels or other physiological states induced by experimental manipulation could result in changes in social hierarchy. This relationship may be mediated through distinct neural microcircuits or receptor mechanisms, emphasizing the need for further research to elucidate the circuitry and synaptic mechanisms underpinning the intrinsic connections between social hierarchy and anxiety levels in mice.

It is important to highlight that optogenetic activation of the SNr[Glu] neurons and SNr[Glu]-DRN pathway produced distinct effects on anxiety, specifically effect-free responses versus alleviation. This discrepancy may stem from several factors. First, numerous studies have demonstrated the role of the DRN in modulating anxiety levels in mice [85–88]. Second, it is widely recognized that stimulation of the axonal terminals of a projection selectively influences the activity within that pathway and its downstream targets [89–91]. Accordingly, it is reasonable to speculate that activation of the SNr[Glu]-DRN pathway may engage downstream neural circuits responsible for regulating anxiety-related behaviors through glutamate release. However, the specific types of DRN neurons that form synaptic connections with SNr glutamatergic neurons require further investigation. Finally, the SNr[Glu] neurons projects to several brain regions as identified in the current findings, including the CPu, VPM, ZID, and DRN. Optogenetic activation of SNr[Glu] neurons may influence multiple downstream targets. Consequently, the absence of anxiety modulation following this activation may result from the ineffectiveness of these neurons or potential counteractions among various pathways. Future research should aim to elucidate the underlying mechanisms contributing to this phenomenon.

In summary, our study indicates that the activation of SNr glutamatergic neurons is both necessary and sufficient for the rapid induction of winning in social competitions. Notably, this effect is not mediated by an enhancement of aggression or physical strength; rather, it stems from the initiation and maintenance of effortful behaviors associated with social competition. These findings carry significant implications for future research on social behavior and could inform the development of therapeutic interventions for neuropsychiatric disorders, including anxiety. Future studies employing more refined manipulations of SNr glutamatergic neurons and their downstream targets are anticipated to yield a deeper understanding of the neural circuitry that regulates social hierarchy and anxiety. For instance, the current findings suggest that optogenetic activation of the SNr[Glu]-DRN pathway is implicated in the regulation of social hierarchy, however, further research is required to comprehensively elucidate the roles of this pathway, utilizing more advanced methodologies such as axon-specific calcium imaging with presynaptically targeted GCaMPs [92]. Additionally, although we ensured that the subjects were thoroughly acclimated to the experimental environment prior to testing, we cannot entirely rule out the possibility that the tube test itself may elicit anxiety-related behavioral changes (e.g., immobility) in mice. Consequently, we are unable to fully ascertain how such alterations may impact their performance during the test. Therefore, further investigation is imperative to elucidate the relationship between the tube test, anxiety levels, and animal performance, underscoring the need for additional experimental studies.

## Materials and methods

### Ethics statement

The care and use of animals, as well as the experimental protocols employed in this study, were conducted in accordance with the guidelines established by the Animal Care and Use Committee of the Chengdu Institute of Biology, Chinese Academy of Sciences (permit number: CIBDWLL2020001), adhering to the Regulations on the Administration of Laboratory Animals (2017 Revision) in China.

### Animals

In the present study, adult male C57BL/6 mice (aged 8–12 weeks) were obtained from Chengdu Dossy Experimental Animals Co., (Chengdu, China). The mice were housed under controlled experimental conditions, featuring a 12-hour light/dark cycle, a temperature maintained at $22 \pm 1$ °C, and a humidity level of $65 \pm 5\%$, with ad libitum access to food and water. The mice were randomly assigned to experimental and control groups. Although a specific statistical method was not employed to predetermine the sample size, the sizes utilized were consistent with those conventionally accepted in the field.

### Surgery

Mice were anesthetized using 1%–2% isoflurane in oxygen and positioned in a stereotaxic apparatus (51603, Stoelting, USA) equipped with a heating pad. Anesthesia depth was consistently maintained at a stable level, indicated by the absence of reflexes. Viral vector microinjection was conducted with a microliter syringe (NRS75 RN 5.0 μL (33/20/3), Hamilton, Switzerland) attached to a syringe pump (Legato 130, KD Scientific, USA) securely mounted to the stereotaxic apparatus. A volume ranging from 200 to 500 nl of AAV solution was administered into the SNr (AP = −3.3 mm, ML = ± 1.5 mm, DV = −4.5 mm). Following the injection, the micropipette was held in place for 10 min before being gently retracted. We used the following high titer AAVs (in genomic particles/ml): AAV2/9-CaMKIIα-GCaMP6f ($2.94 \times 10^{12}$, 400 nl/site for optical fiber-based Ca$^{2+}$ recordings), AAV2/9-CaMKIIα-hChR2(H134R)-EYFP ($2.86 \times 10^{12}$; 300 nl/site for optogenetic activation), AAV2/9-CaMKIIα-hM4D(Gi)-EGFP ($4.89 \times 10^{12}$; 300 nl/site for chemogenetic inhibiton) and AAV2/9-CaMKIIα-EYFP ($2.73 \times 10^{12}$; 200 nl/site for anterograde tracing and control groups). Two weeks post-viral injection, two fiber optic cannulas (1.25 mm in diameter; Fiblaser, Shanghai, China) were implanted above the SNr (AP = −3.3 mm, ML = ± 1.5 mm, DV = −4.0 mm) or the DRN (AP = −5.5 mm, ML = ± 0 mm, DV = −2.0 mm; at a 15° angle from caudal to rostral) for subsequent optogenetic activation and fiber photometry experiments, respectively. Post-surgery, the mice were maintained on a heating pad until they regained ambulatory function, after which they were returned to their home cages. Behavioral experiments were conducted 3 weeks later to allow for sufficient viral expression.

### Behavioral tests

**Tube test.** The tube test, a well-established methodology for evaluating social dominance among mice, has been extensively documented in previous research [93]. In brief, following a co-housing period of at least 2 weeks in the same cage, which included a 15 cm-long acrylic tube with an inner diameter of 3 cm for animal acclimatization, four adult male C57BL/6 mice underwent three consecutive days of handling, succeeded by three consecutive days of tube test training. Training utilized an acrylic tube with an internal diameter of 3 cm and a length of 30 cm, with each mouse completing 10 tube entries per day (5 entries from each end). During the tube test, pairs of mice were placed at contrary ends of the tube, converging in the middle. The mouse that exited the tube first was designated as the loser, while its counterpart was labeled as the winner. Should neither mouse exit the tube within a 2-min period during a trial, the test was repeated. Each mouse engaged in three trials daily against its cage mates, culminating in a total of six trials per cage. The rankings of the four mice were subsequently determined based on the total number of victories achieved.

The behaviors of the animals during the tube test were meticulously captured on video and subsequently analyzed frame by frame using BORIS video analysis software (V7.12.2, University of Torino) [94]. In accordance with the behavioral classification criteria established in a previous study, the observed actions within the tube were categorized as follows: push-initiation (actively instigating a push), push-back (responding with a push after being challenged by an opponent), resistance (withstanding backward movement when confronted), stillness (exhibiting no movement beyond sniffing and grooming), and retreat (receding backward either in response to being pushed or by voluntary decision). Each distinct behavioral epoch was carefully annotated using BORIS to facilitate further analysis. To eliminate any potential bias in the analysis, the annotator was blinded to the hierarchical ranks or prior experiences of the mice during the behavioral annotation process.

**Push ball test.** To differentiate social rank-related efforts from nonsocial physical exertion, we conducted the push ball test. The experimental procedure for this test closely resembled that of the tube test, with the following modifications: a 15 cm-long acrylic tube with an inner diameter of 3 cm was provided, along with a 2.5 cm-diameter diamond-shaped rubber bouncy ball, both placed in the cage 1 week in advance to allow for animal acclimatization. Furthermore, the push ball test utilized a 45 cm-long tube, with the ball positioned 15 cm from one end. The mouse entered from this end and pushed the ball forward through the tube (the ball travels a distance of 30 cm) to complete one trial. A total of four trials were conducted (two entries from each end). The moment the mouse initiated ball pushing was designated as event time 0 for $Ca^{2+}$ signal analysis. Additionally, we analyzed the number of active pushes and the latency required to displace the ball from the tube under both light-stimulated and nonstimulated conditions for optogenetic manipulation.

**Warm spot test.** An adequate volume of water was poured into a rectangular plastic container (28 cm × 20 cm × 20 cm), which was subsequently placed in a freezer. Prior to the initiation of the experiment, a plastic film was affixed to the surface of the ice to maintain the temperature at approximately 0 ℃ throughout the duration of the test. A heating coil, encased in cardboard, was positioned in one corner of the container to elevate the local temperature to 34 ℃. This configuration was designed to simulate a warm nest with a diameter of 5 cm, accommodating a single adult mouse only. For the preparatory phase, four mice from a single cage were initially housed in a container with a base layer of ice, devoid of a warm nest, for 20 min to facilitate a reduction in their body temperature. Following this acclimatization period, the mice were transferred to the experimental container, equipped with both ice and a warm nest. The competition among the four mice for access to the warm nest was subsequently observed and recorded over a duration of 20 min.

The warm spot test was conducted on both hM4Di-EGFP mice and EYFP mice. One day following the initial test, the selected individual received an intraperitoneal injection of 5 mg/kg CNO, while its cage mates were administered an equivalent volume of saline solution. The warm spot test was subsequently repeated at 2 and 48 hours post-CNO injection. Using the BORIS software, we annotated the timestamps of entries into and evictions from the warm nest, and calculated the total duration each mouse occupied the nest. In instances where multiple animals occupied the warm nest simultaneously, only the mouse occupying the majority of the area within the warm nest was recorded. As with prior procedures, the annotator was blinded to the hierarchical ranks and past experiences of the mice during the behavioral annotation process.

**OFT.** To evaluate the motivation for exploring the central area of a novel environment and to monitor the locomotor activity of the test subjects, we gently placed each mouse in the center of a 40 × 40 cm open field apparatus and recorded its activity under dim light during the dark phase for a duration of 10 min. The open field test was conducted using a behavioral research system (CinePlex, Plexon, USA) for video recording and data acquisition. In optogenetic experiments, blue light (473 nm, 20 Hz, 20 ms, 10 mW) was intermittently activated for 1-min epochs. For chemogenetic experiments, the open field test was performed 2 hours post-CNO injection. The tracking software (CinePlex Studio V3.5.0) was employed to calculate the number of entries into the central zone, the duration of stay in the central zone (center time), and the total distance traveled (locomotion) for each minute.

**EPM.** Elevated plus-maze assays were conducted to evaluate anxiety-like behaviors in mice utilizing a four-arm maze configuration comprising two open and two closed arms. Each mouse was placed in the central intersection of the maze, and their locomotor activities were recorded using the previously described system under low-light conditions during the dark phase over a 9-min session. The duration spent in both open and closed arms was quantified using the CinePlex Studio software. In experimental groups, blue light (473 nm, 20 Hz, 20 ms, 10 mW) was intermittently activated for 3-min intervals.

**Social memory test.** The social memory test was performed using a three-chamber apparatus (40×20 cm). Each test mouse was allowed to habituate in the central chamber for 10 min. Subsequently, an unfamiliar mouse was introduced into one of the side chambers. Following a 5-min familiarization period with the introduced mouse, a novel object mouse was placed in the other side chamber. The subsequent 5-min interval was used to observe the exploration behaviors of the test mice towards either the familiar or novel object mouse. For analysis, the time spent in the interaction zone (defined as half of the center chamber adjacent to the side chamber, 10×20 cm) was recorded in 1-min epoch, during which the mice experienced alternating light-on and light-off periods (473 nm, 20 Hz, 20 ms, 10 mW, activated and deactivated intermittently in 1-min epochs).

**Social interaction test.** To distinguish social rank-related efforts from social, noncompetitive physical exertion, we performed the social interaction test. The test was conducted using a procedure analogous to that of the social memory test. An 18 cm-high and 10 cm-diameter restraint cage was positioned on a platform (4 cm-high and 10 cm in diameter) at the midpoint of the long side of the test chamber (40 cm×20 cm, devoid of partitions). Each test mouse was allowed to acclimate in the chamber for 10 min prior to the introduction of a novel mouse into the restraint cage. During the subsequent 5-min session, the test mouse was free to approach and explore the novel mouse within the cage. For fiber photometry recording, the analysis of calcium signals was anchored to two distinct event time points: the initiation of the test mouse's movement toward the novel mouse and the onset of lifting the head to sniff or forelimb elevation in preparation for contacting the lower edge of the restraint cage. In the context of optogenetic manipulation, analyses concentrated on the duration of specific behaviors, which included voluntary approach (defined as the period from the moment the test mouse began to move toward the novel mouse until it reached the restraint cage), effortful social interaction (encompassing head lifting to sniff or raising the forelimb to touch the lower edge of the restraint cage for sniffing), nonsocial exploration (involving exploration of areas beyond the restraint cage), and overall social behavior (comprising both voluntary approach and effortful social interaction).

**Resident-intruder test.** The standard resident-intruder test was conducted on mice that had cohabitated with their cage mates for a duration exceeding one month. Prior to the assay, all cage mates were removed from the home cages, allowing the test mice to habituate for 10 min. Subsequently, adolescent male intruders (5 weeks old) were introduced into the cages for a 10-min testing period. Concurrently, blue light (473 nm, 20 Hz, 20 ms, and 10 mW) was intermittently administered in 1-min epochs on the test mice. The frequency and duration of aggressive behaviors (e.g., chasing and attacking) and nonaggressive social interactions (e.g., sniffing and social grooming) exhibited by the test mice were recorded and analyzed for each 1-min epoch corresponding to the light-on and light-off intervals.

**Grip strength measurement.** We employed a force gauge (DS2-50N, Puyan, Dongguan, China) to measure the forelimb grip strength of mice. Each mouse underwent three testing blocks with 2-hour intervals between blocks. Within each block, mice completed 10 trials over approximately 6 min, comprising two equal 3-min periods corresponding to light-on or light-off conditions (473 nm, 20 Hz, 20 ms, and 10 mW). The sequence of light-on and light-off periods was varied pseudorandomly across the three blocks to mitigate potential order biases. Ultimately, the maximal grip strengths recorded from the three blocks under both light-on and light-off conditions were used for statistical analysis.

**Pole test.** The pole test assesses the grasping abilities of the forelimbs and overall motor coordination in mice. In this assay, mice were positioned head-up atop an upright wooden pole, measuring 50 cm in height and 1 cm in diameter, which was securely attached to a square base. Prior to the experiment, the mice were acclimated in the testing room for 15 min.

We recorded the duration required for each mouse to rotate from a head-up to a head-down position and subsequently descend from the top to the bottom of the pole. Each mouse underwent a 2-day training regimen, consisting of five trials per day. On the third day, the mice were subjected to three test trials, with the pole being meticulously cleaned with alcohol after each individual test. Averages were calculated for subsequent statistical analyses.

**Rotarod test.** The rotarod test was conducted to evaluate locomotor activity and coordination in mice. In this assay, mice were required to traverse a rotating cylindrical rod. Prior to the experiment, the mice were acclimated in the testing room for 15 min. Subsequently, a 2-day training regimen was implemented, during which the mice underwent training on the rotarod apparatus (XR-6C, Xinruan, Shanghai, China) at a low, constant speed of 5 rpm until they could maintain stability on the rod for 5 min without falling. On the third day, a constant acceleration mode (ranging from 4 to 40 rpm with an acceleration time of 5 min) was employed during the test. Mice were required to coordinate their movements and maintain balance to avoid falling from the rod. In the event of a fall, latency, speed, and total distance traveled during the trial were automatically recorded. Each mouse completed three trials, with a minimum interval of 15 min between trials. The rod was thoroughly cleaned with alcohol after each individual test, and averages were subsequently calculated for statistical analysis.

## Optogenetic manipulation of SNr and its downstream targets in the tube test

Following the establishment and maintenance of a stable rank in the tube test for over 3 days, all four mice were fitted with simulated optic fibers and underwent preliminary tube tests (featuring a 12 mm slit at the top of the tube) for two consecutive days prior to the main experiments. Photostimulation for each test subject commenced at 5 mW, with the tube test performed against the cage mate closest in rank, subsequently progressing to opponents with increasingly greater rank disparities. Mice were required to win or lose in two out of three trials from either end of the tube. If the test mice secured victories from both ends of the tube, it was inferred that photostimulation at that specific intensity successfully induced a rank alteration. Conversely, if no rank change occurred, the light intensity was incrementally increased, and the procedure was repeated until a rank shift was observed or the light intensity reached 20 mW. The total number of winning trials induced by photostimulation in the test mouse was recorded. Subsequently, rank changes for each mouse were subjected to detailed statistical analyses, with rank 1 mice excluded due to the ceiling effect.

## Chemogenetic manipulation of SNr in the tube test

Male mice utilize chemosensory signals to assess the dominance status of their opponents during social interactions [95]. A preliminary experiment revealed that optogenetic inhibition of SNr$^{Glu}$ neurons led to increased periods of immobility and reduced active competitive behavior in the test subject; however, the opponent ultimately retreated, potentially due to the winner effect or a lack of chemosensory signals reflecting the internal state changes in the test subject. Consistent with numerous studies employing both optogenetics and chemogenetics to bidirectionally modulate relevant brain regions and their associated circuits [25,27,68], we also utilized chemogenetic methods for inhibition. This choice was informed by the possibility that, during the intervals between tests, the subject's chemosensory cues may change and disperse (e.g., via urine) within the housing cage. All four mice housed in the same cage were administered the rAAV-CaMKIIα-hM4D(Gi)-EGFP-WPRE-hGH-pA virus via injection. At least 4 weeks post-viral injection, the tube test was conducted at intervals of 1–1.5, 3–5, 6–8, 24, 48, and 72 hours following an intraperitoneal (i.p.) saline injection. One week later, the test mice received an i.p. injection of CNO (5 mg/kg), while the remaining mice were given saline injections. The tube test was then repeated at the specified time points. Test subjects were selected randomly. Following these procedures, rank changes for each mouse were evaluated through comprehensive statistical analyses, excluding rank 4 mice due to the floor effect.

## Fiber photometry recording and analysis

Ca$^{2+}$ signals were recorded utilizing a fiber photometry system (QAXK-FPS-SS-LED, ThinkerTech, Nanjing, China). The GCaMP6f indicator was excited with 488 nm light, and the signals at 405 nm were subtracted from those at 488 nm to

obtain corrected signals. Data were sampled at a frequency of 100 Hz, and the laser intensity was maintained at 40 μw at the tip of the optic fiber to minimize photobleaching. Behavioral observations during the tube test were captured using a camera positioned beside the tube and subsequently annotated with BORIS for further analysis. To synchronize the fiber photometry recordings with the video, an external trigger generated a TTL pulse to the recording system while simultaneously illuminating a red light-emitting diode, which was captured by the camera. Each pair of opponents underwent six trials, consisting of three entries from each side. As a control, mice were allowed to traverse the tube alone in six trials, with three entries from each side. To ensure effective signal acquisition in each mouse and to mitigate possible confounding effects associated with potential lateralization of the SNr, calcium signals were recorded from the left SNr in half of the randomly selected subjects, while the remaining subjects had signals recorded from the right counterpart.

$Ca^{2+}$ signals were analyzed using proprietary scripts provided by ThinkerTech Nanjing Biotech Co., which were developed in MATLAB (2017b). Fluorescence changes, denoted as ΔF/F0, were calculated using the formula (F-F0)/F0, where F0 represents the baseline fluorescence averaged over a 2-s interval preceding the onset of a behavioral epoch. For peri-event time histogram (PETH) analysis, behavioral onset was aligned to time zero, and fluorescence signals were normalized using Z-score and binned at 10 ms intervals. Specifically, for the PETH analysis of mice traversing a tube, fluorescence signals were aligned to the moment that the mice reached the midpoint of the tube. Statistical significance of behavior-associated fluorescence changes was evaluated using a permutation test, following previously established methodologies.

## Histology and immunohistochemistry

Mice were anesthetized using 1% pentobarbital sodium and subsequently perfused transcardially, initially with saline and then with 4% paraformaldehyde. The brains were excised and fixed in 4% paraformaldehyde for a period of 24 hours, followed by storage in a 30% sucrose/phosphate buffer solution (PBS) at 4 °C until sectioning. Coronal brain slices of 30 μm thickness were prepared for immunohistochemical staining using a cryostat (CM1850, Leica, Germany). Primary antibodies used in this study included rabbit anti-CaMKIIα (1:300; Abcam, ab5683), rabbit anti-VGLUT2 (1:500; Abcam, ab216463), and rabbit anti-TH (1:500, Millipore, AB152). Secondary antibody employed was donkey anti-rabbit Alexa Fluor 647 (1:500, Abcam, ab150111). The sections were subsequently mounted onto slides, coverslipped, and imaged using a fluorescence microscope (Nexcope NE910, Ningbo, China). Brain regions were identified according to the Mouse Brain Atlas [96], and the areas designated for quantification were delineated by outlining the boundaries of the SNr. Cell counting was conducted using Fiji software [97].

## Statistical analysis and Reproducibility

The normality of distributions and the homogeneity of variance for all data were assessed using the Shapiro–Wilk test and Levene's test, respectively. In the tube test, the Wilcoxon signed rank test was utilized to assess differences in behavioral parameters across various light conditions in optogenetic experiments, while the Mann–Whitney $U$ test was employed to compare these parameters among different groups in chemogenetic experiments. For the warm spot test, one-way repeated measures ANOVA with Bonferroni correction was used to compare the durations of occupying the warm spot, while the Friedman test was applied to compare ranks across different time points. For the open field test, one-way repeated measures ANOVA with Bonferroni correction was utilized to compare differences in behavioral parameters among groups subjected to chemogenetic manipulation, and paired-samples $t$ tests were conducted to compare behavioral parameters across different light conditions for optogenetic manipulation. For the elevated plus-maze and resident-intruder tests, two-way repeated measures ANOVA with Bonferroni correction was employed, with the variables being "light condition" and "group". For the social memory test, two-way repeated measures ANOVA with Bonferroni correction was used, considering the variables "zone" and "group". For the push ball and social interaction tests, two-way repeated measures ANOVA with Bonferroni correction was employed, considering the variables of "light condition" and "group".

Similarly, for the pole and rotarod tests, two-way repeated measures ANOVA with Bonferroni correction was utilized, focusing on the variables of "injection (CNO vs. saline)" and "group". Paired-samples $t$ tests were conducted to compare forelimb grip strength between different light conditions. Statistical analyses were performed using SPSS software (release 23.0) with a significance threshold set at $p < 0.05$ and a highly significant threshold at $p < 0.001$. The reproducibility of micrographic and behavioral experiments in optogenetics, chemogenetics, and fiber photometry recordings was consistent with the actual number of experimental subjects, and the repeatability of micrograph statistics corresponded to the sample size. Furthermore, the repeatability of anterograde tracing experiments was confirmed to be at least three times. The dataset generated and analyzed during the current study is available in the Dryad repository [98].

## Supporting information

**S1 Fig. Verification of viral expression in the SNr.** (A) Histological data from four individual animals, with each column representing a separate subject. The left side of each column displays a representative coronal section of the SNr illustrating the expression of EYFP; the right pie presents the percentage of SNr neurons expressing the virus relative to the total number of virus-expressing neurons in the entire field of view. (B–D) Similar analyses for GCaMP6, ChR2-EYFP, and hM4Di-EGFP expression, respectively. Scale bar, 200 μm.
(PDF)

**S2 Fig. Immunohistological images demonstrating colocalization of virus and CaMKIIα+ neurons in the SNr.** (A) Immunohistological data from two individual subjects, with each row representing a distinct animal. The left three subplots display the expression of EYFP+, CaMKIIα+, and the colocalization of EYFP+ and CaMKIIα+ within the SNr, respectively. The right pie illustrates the percentage of CaMKIIα+ neurons co-labeled with EYFP in the SNr. (B–D) Corresponding analyses for GCaMP6, ChR2-EYFP, and hM4Di-EGFP expression, respectively. Scale bar, 100 μm.
(PDF)

**S3 Fig. Immunohistological images demonstrating colocalization of virus and VGLUT2+ neurons in the SNr.** (A) The left three subplots display the expression of EYFP+, VGLUT2+, and the colocalization of EYFP+ and VGLUT2+ within the SNr, respectively. The right pie illustrates the percentage of VGLUT2+ neurons co-labeled with EYFP in the SNr. (B–D) Similar analyses for GCaMP6, ChR2-EYFP, and hM4Di-EGFP expression, respectively. Scale bar, 60 μm.
(PDF)

**S4 Fig. $Ca^{2+}$ Activity of $SNr^{Glu}$ neurons during pushes in the push ball test. (A)** $Ca^{2+}$ signals were aligned with the onset of pushes when mice expressing GCaMP6f pushed the diamond-shaped rubber bouncy ball within the tube. The upper panel illustrates a heatmap of $Ca^{2+}$ signals aligned to the push-initiation, with each row representing an individual trial ($n = 121$ trials from 8 mice). The middle panel depicts a heatmap displaying the average $Ca^{2+}$ signals for each animal across all trials ($n = 8$). The lower panel presents the mean $Ca^{2+}$ transients associated with pushes for the entire test group ($n = 8$), where solid lines indicate the mean and shaded areas represent SEM. No significant changes in calcium signals associated with pushes were observed in GCaMP6f-expressing mice ($p > 0.05$; permutation test). (**B**) $Ca^{2+}$ signals were aligned with the initiation of pushes when mice expressing EYFP encountered the ball within the tube. The upper panel presents data from all trials ($n = 84$ trials from 6 mice), the middle panel shows data from all animals ($n = 6$), and the lower panel displays the mean ± SEM of the average $Ca^{2+}$ signals. No significant changes in calcium signals associated with pushes were observed in EYFP-expressing mice ($p > 0.05$; permutation test). The data underlying this Figure can be found in files numbered 19–20 on Dryad (https://doi.org/10.5061/dryad.m0cfxppg3).
(PDF)

**S5 Fig. $Ca^{2+}$ Activity of $SNr^{Glu}$ neurons during voluntary approach and effortful social interaction in the social interaction test. (A)** $Ca^{2+}$ signals were aligned with the initiation of voluntary approach when mice expressing GCaMP6f

encountered a novel mouse introduced into the restraint cage. The upper panel presents a heatmap of $Ca^{2+}$ signals aligned to the onset of voluntary approach, with each row representing an individual trial ($n = 54$ trials from 8 mice). The middle panel displays a heatmap depicting the average $Ca^{2+}$ signals for each animal across all trials ($n = 8$). The lower panel illustrates the mean $Ca^{2+}$ transients associated with voluntary approach for the entire test group ($n = 8$), where solid lines indicate the mean and shaded areas denote SEM. No significant changes in calcium signals associated with voluntary approach were observed in GCaMP6f-expressing mice ($p > 0.05$; permutation test). **(B)** $Ca^{2+}$ signals were aligned with the onset of effortful social interaction when mice expressing GCaMP6f encountered the novel mouse in the restraint cage. The upper panel presents data from all trials ($n = 98$ trials from 8 mice), the middle panel summarizes data from all animals ($n = 8$), and the lower panel shows the mean $\pm$ SEM of the average $Ca^{2+}$ signals. No significant changes in calcium signals associated with effortful social interaction were observed ($p > 0.05$; permutation test). Panels **(C)** and **(D)** replicate the findings presented in (A) and (B), respectively, with the only distinction being that (C) and (D) pertain to EYFP-expressing mice, involving 66 trials from 6 animals for (C) and 83 trials from 6 animals for (D). No significant changes in calcium signals were detected for EYFP-expressing mice ($p > 0.05$; permutation test). The data underlying this Figure can be found in files numbered 21–24 on Dryad (https://doi.org/10.5061/dryad.m0cfxppg3).
(PDF)

**S6 Fig. Photostimulation of SNr^Glu neurons did not affect behaviors and ranks of EYFP mice in the tube test.** **(A)** Number and duration of push-initiation for each animal in the tube test ($n = 8$, Wilcoxon signed rank test; number: $Z = -0.845$, $p = 0.398$; duration: $Z = 0.000$, $p = 1.000$). **(B)** Number and duration of push-back for each animal in the tube test ($n = 8$, Wilcoxon signed rank test; number: $Z = -0.141$, $p = 0.888$; duration: $Z = -1.400$, $p = 0.161$). **(C)** Number and duration of stillness for each animal in the tube test ($n = 8$, Wilcoxon signed rank test; number: $Z = -0.281$, $p = 0.779$; duration: $Z = -1.680$, $p = 0.093$). **(D)** Percentage of time spent resisting ($n = 8$, Wilcoxon signed rank test; $Z = -0.700$, $p = 0.484$). **(E)** Percentage of time spent retreating ($n = 8$, Wilcoxon signed rank test; $Z = -0.980$, $p = 0.327$). **(F)** Example of rank positions for one cage of mice tested daily over 7 days, showing that the third-ranked mouse did not change its rank following photostimulation of SNr^Glu neurons. **(G)** Summary of rank changes in EYFP mice before and after photostimulation on SNr^Glu neurons ($n = 9$). Each line represents one animal. The data underlying this Figure can be found in file number 25 on Dryad (https://doi.org/10.5061/dryad.m0cfxppg3).
(PDF)

**S7 Fig. Optogenetic activation of SNr^Glu neurons did not impact anxiety levels, social preference, aggressive behavior, grip strength of mice, and physical effort devoid of social competition. (A)** During the elevated plus-maze (EPM) test, time spent in the open arm decreased over time ($n = 6$ for ChR2-EYFP mice and $n = 7$ for EYFP mice, two-way repeated measures ANOVA with Bonferroni correction; $F_{2,22} = 15.814$, $p < 0.001$), though no significant differences were observed between groups ($F_{1,11} = 0.621$, $p = 0.447$). **(B)** In the social memory test, both ChR2-EYFP and EYFP mice showed a significant preference for novel mice during light-on period ($n = 8$ and 10 for ChR2-EYFP and EYFP mice, respectively; two-way repeated measures ANOVA with Bonferroni correction; $F_{1,16} = 11.293$, $p = 0.004$). However, no significant differences were found between groups ($F_{1,16} = 4.127$, $p = 0.059$). **(C)** The resident-intruder test indicated that both ChR2-EYFP and EYFP mice displayed normal levels of aggressive and nonaggressive behaviors towards novel mice ($n = 8$ for each group, two-way repeated measures ANOVA with Bonferroni correction; $F_{1,14} = 0.146$, $p = 0.708$), with no significant differences between groups ($F_{1,14} = 0.137$, $p = 0.716$). **(D)** Optogenetic activation of SNr^Glu neurons did not affect forelimb grip strength during light-on and light-off periods ($n = 8$, paired two-sided $t$ test; $t_7 = 1.735$, $p = 0.126$). **(E)** In the push ball test, the analysis of latency revealed no significant main effects for "light condition" ($n = 7$ for each group; two-way repeated measures ANOVA with Bonferroni correction; $F_{1,12} = 0.131$, $p = 0.723$) or "group" ($F_{1,12} = 0.336$, $p = 0.573$). Similarly, the assessment of the number of active pushes showed no significant main effects for "light condition" ($F_{1,12} = 0.220$, $p = 0.647$) or "group" ($F_{1,12} = 0.079$, $p = 0.783$). **(F)** In the social interaction test, the duration analysis indicated

no significant main effects for "light condition" ($n = 7$ for each group; two-way repeated measures ANOVA with Bonferroni correction): voluntary approach ($F_{1,12} = 1.700$, $p = 0.217$), effortful social interaction ($F_{1,12} = 0.017$, $p = 0.897$), overall social behavior ($F_{1,12} = 0.696$, $p = 0.421$), and nonsocial exploration ($F_{1,12} = 0.314$, $p = 0.586$). Furthermore, no significant main effects were observed for "group" in the same parameters: voluntary approach ($F_{1,12} = 0.064$, $p = 0.804$), effortful social interaction ($F_{1,12} = 0.549$, $p = 0.473$), overall social behavior ($F_{1,12} = 0.551$, $p = 0.472$), and nonsocial exploration ($F_{1,12} = 0.128$, $p = 0.727$). *$p < 0.05$; **$p < 0.01$. The data underlying this Figure can be found in files numbered 26–35 on Dryad (https://doi.org/10.5061/dryad.m0cfxppg3).
(PDF)

**S8 Fig. Chemogenetic inhibition of SNr^Glu neurons in hM4Di-EGFP mice reduces effortful behaviors in the tube test 3.5 hours post-CNO injection compared to saline injection. (A)** Number and duration of push-initiation for each animal in the tube test ($n = 7$ mice for saline injection and $n = 9$ mice for CNO injection, Mann–Whitney $U$ test; number: U = 10, $p = 0.011$; duration: U = 5, $p = 0.005$). **(B)** Number and duration of push-back for each animal in the tube test ($n = 7$ versus 9, Mann–Whitney $U$ test; number: U = 20, $p = 0.218$; duration: U = 26, $p = 0.560$). **(C)** Number and duration of stillness for each animal in the tube test ($n = 7$ versus 9, Mann–Whitney $U$ test; number: U = 18, $p = 0.140$; duration: U = 13, $p = 0.050$). **(D)** Percentage of time spent resisting ($n = 7$ versus 9, Mann–Whitney $U$ test; U = 12, $p = 0.039$). **(E)** Percentage of time spent retreating ($n = 7$ versus 9, Mann–Whitney $U$ test; U = 7, $p = 0.004$). Note that the same 3 animals were used in both groups; specifically, 3 out of 9 mice received i.p. injection of saline 1 week prior to the CNO injection. * $p < 0.05$; ** $p < 0.01$. The data underlying this Figure can be found in file number 36 on Dryad (https://doi.org/10.5061/dryad.m0cfxppg3).
(PDF)

**S9 Fig. Chemogenetic manipulation of SNr^Glu neurons in EYFP mice did not influence the duration of occupation or ranking in the warm spot test following injections of either saline or CNO. (A)** The duration (top) and rank (bottom) in the warm spot test were assessed for mice expressing EYFP in the SNr at 1 day before, 2 hours post, and 2 days post i.p. administration of saline (duration in the warm spot: $n = 9$, one-way repeated measures ANOVA with Bonferroni correction, $F_{2,16} = 0.007$; $p = 0.993$; rank: Friedman test, $t_2 = 0.000$; $p = 1.000$). **(B)** Similarly, the duration (top) and rank (bottom) in the warm spot test for mice expressing EYFP in the SNr at 1 day before, 2 hours after, and 2 days following i.p. injection of CNO (duration: $n = 8$, one-way repeated measures ANOVA with Bonferroni correction, $F_{2,14} = 2.273$; $p = 0.140$; rank: Friedman test, $t_2 = 0.667$; $p = 0.717$). Notably, all mice received an i.p. injection of saline 1 week prior to the CNO injection. Each line on the graph represents an individual animal. The data underlying this Figure can be found in files numbered 37–38 on Dryad (https://doi.org/10.5061/dryad.m0cfxppg3).
(PDF)

**S10 Fig. Chemogenetic manipulation of SNr^Glu neurons did not affect locomotor ability or coordination in mice, as assessed by the pole test (A and B) and rotarod test (C–E) 2 hours post-injection of either CNO or saline. (A)** There was no significant difference in the duration required for mice to transition from the head-up to head-down position between CNO and saline injections ($n = 9$ for both hM4Di-EGFP and EYFP groups; two-way repeated measures ANOVA with Bonferroni correction; $F_{1,16} = 0.067$, $p = 0.799$) or between hM4Di-EGFP and EYFP mice ($F_{1,16} = 0.013$, $p = 0.910$). **(B)** The total duration from head-up to descent at the bottom of the pole did not significantly differ between CNO and saline injections ($F_{1,16} = 0.007$, $p = 0.936$) or between hM4Di-EGFP and EYFP mice ($F_{1,16} = 0.377$, $p = 0.548$). **(C)** No significant difference in latency to fall was observed between CNO and saline injections ($n = 8$ for both hM4Di-EGFP and EYFP groups; two-way repeated measures ANOVA with Bonferroni correction; $F_{1,14} = 0.163$, $p = 0.692$) or between hM4Di-EGFP and EYFP mice ($F_{1,14} = 0.211$, $p = 0.653$). **(D)** Distance traveled did not differ significantly between CNO and saline injections ($F_{1,14} = 0.137$, $p = 0.717$) or between hM4Di-EGFP and EYFP mice ($F_{1,14} = 0.163$, $p = 0.693$). **(E)** There was also no

significant difference in speed at the moment of fall between CNO and saline injections ($F_{1,14} = 0.705$, $p = 0.415$) or between hM4Di-EGFP and EYFP mice ($F_{1,14} = 0.990$, $p = 0.337$). The data underlying this Figure can be found in files numbered 39–43 on Dryad (https://doi.org/10.5061/dryad.m0cfxppg3).
(PDF)

**S11 Fig. The optogenetic stimulation of the SNr<sup>Glu</sup>-CPu, -VPM, and -ZID pathways did not influence the social hierarchy of mice.** Panels **(A)**, **(C)**, and **(E)** illustrate diagrams of the unilateral viral infection area using rAAV-CaMKIIα-hChR2(H134R)-EYFP-WPRE-hGH-pA in the SNr, along with the placement of optic fibers above the CPu **(A)**, VPM **(C)**, and ZID **(E)**, respectively. Panels **(B)**, **(D)**, and **(F)** present examples of rank positions for a group of mice assessed daily over 7 days, showing that the rank of the third-position mouse remained unchanged following photostimulation of SNr<sup>Glu</sup> neuronal terminals in the CPu (B), VPM (D), and ZID (F), respectively. Each line represents an individual animal. Abbreviations: CPu, caudate putamen (striatum); VPM, ventral posteromedial thalamic nucleus; ZID, zona incerta, dorsal part. The stereotaxic coordinates (in mm) are: CPu (AP = 0.5, ML = ± 2.0, DV = −3.0), VPM (AP = −1.0, ML = ± 0.8, DV = −3.5), and ZID (AP = −1.8, ML = ± 1.0, DV = −4.0).
(PDF)

**S12 Fig. Photostimulation of the SNr<sup>Glu</sup>-DRN pathway did not significantly alter the behaviors or hierarchical ranks of EYFP mice. (A)** Number and duration of push-initiation for each animal in the tube test ($n = 7$, Wilcoxon signed rank test; number: $Z = −0.594$, $p = 0.553$; duration: $Z = −0.000$, $p = 1.000$). **(B)** Number and duration of push-back for each animal in the tube test ($n = 7$, Wilcoxon signed rank test; number: $Z = −0.338$, $p = 0.735$; duration: $Z = −0.676$, $p = 0.499$). **(C)** Number and duration of stillness for each animal in the tube test ($n = 7$, Wilcoxon signed rank test; number: $Z = −0.508$, $p = 0.611$; duration: $Z = −1.014$, $p = 0.310$). **(D)** Percentage of time spent resisting ($n = 7$, Wilcoxon signed rank test; $Z = −0.845$, $p = 0.398$). **(E)** Percentage of time spent retreating ($n = 7$, Wilcoxon signed rank test; $Z = −0.338$, $p = 0.735$). **(F)** Daily rank positions of a sample cage of mice over 7 days, illustrating that the third-ranked mouse maintained its rank following photostimulation of SNr<sup>Glu</sup> neuron terminals in the DRN. **(G)** No changes in rank were observed in EYFP mice before and after the optogenetic activation of the SNr<sup>Glu</sup>-DRN pathway. Each line represents an individual animal. The data underlying this Figure can be found in file number 44 on Dryad (https://doi.org/10.5061/dryad.m0cfxppg3).
(PDF)

**S13 Fig. Photostimulation of the SNr<sup>Glu</sup>-DRN pathway did not influence other behavioral parameters in ChR2-EYFP mice. (A)** Number and duration of stillness for each animal in the tube test for ChR2-EYFP mice ($n = 7$, Wilcoxon signed rank test; number: $Z = −0.677$, $p = 0.498$; duration: $Z = −1.183$, $p = 0.237$). **(B)** Percentage of time spent resisting ($n = 7$, Wilcoxon signed rank test; $Z = −0.676$, $p = 0.499$). **(C)** Both ChR2-EYFP and EYFP mice demonstrated typical levels of aggressive ($n = 8$ for each group, two-way repeated measures ANOVA with Bonferroni correction; $F_{1,14} = 0.173$, $p = 0.684$) and nonaggressive behaviors ($F_{1,14} = 0.207$, $p = 0.656$) in the resident-intruder test, with no significant differences observed between the groups ($F_{1,14} = 0.782$, $p = 0.391$ for aggressive behaviors; $F_{1,14} = 0.06$, $p = 0.811$ for nonaggressive behaviors). **(D)** Optogenetic activation of the SNr<sup>Glu</sup>-DRN pathway did not alter the number of entries into the central area ($n = 7$, paired two-sided $t$ test, $t_6 = 1.307$, $p = 0.239$), the time spent in the center ($n = 7$, paired two-sided $t$ test, $t_6 = 0.941$, $p = 0.383$), or overall locomotion ($n = 7$, Wilcoxon signed rank test, $Z_6 = −1.521$, $p = 0.128$) during the open field test. **(E)** Optogenetic activation of the SNr<sup>Glu</sup>-DRN pathway did not alter forelimb muscle strength during both light-on and light-off periods ($n = 8$, paired two-sided $t$ test; $t_7 = 0.643$, $p = 0.541$). **(F)** In the push ball test, no significant main effects were observed for the number of active pushes with respect to "light condition" ($n = 7$ for each group; two-way repeated measures ANOVA with Bonferroni correction; $F_{1,12} = 0.357$, $p = 0.561$) or "group" ($F_{1,12} = 0.755$, $p = 0.402$). Similarly, analysis of latency revealed no significant main effects for "light condition" ($F_{1,12} = 0.018$, $p = 0.895$) or "group" ($F_{1,12} = 2.703$, $p = 0.126$). **(G)** In the social interaction test, duration analyses indicated no significant main effects for "light condition" ($n = 7$ for each group; two-way repeated measures ANOVA with Bonferroni correction): voluntary approach ($F_{1,12} = 0.636$, $p = 0.441$), effortful social

interaction ($F_{1,12} = 0.009$, $p = 0.927$), overall social behavior ($F_{1,12} = 0.348$, $p = 0.566$), and nonsocial exploration ($F_{1,12} = 0.023$, $p = 0.882$). Additionally, no significant main effects were found for "group" across the same parameters: voluntary approach ($F_{1,12} = 0.063$, $p = 0.807$), effortful social interaction ($F_{1,12} = 0.057$, $p = 0.815$), overall social behavior ($F_{1,12} = 0.279$, $p = 0.607$), and nonsocial exploration ($F_{1,12} = 0.188$, $p = 0.672$). The data underlying this Figure can be found in files numbered 15 and 45–55 on Dryad (https://doi.org/10.5061/dryad.m0cfxppg3).
(PDF)

## Author contributions

**Conceptualization:** Yanzhu Fan, Guangzhan Fang.

**Data curation:** Yanzhu Fan, Shaoxiang Ge, Lidi Lu.

**Formal analysis:** Yanzhu Fan, Shaoxiang Ge.

**Investigation:** Yanzhu Fan, Shaoxiang Ge, Lidi Lu, Wenjun Niu, Zhiyue Wang.

**Methodology:** Yanzhu Fan, Guangzhan Fang.

**Project administration:** Yanzhu Fan, Guangzhan Fang.

**Resources:** Guangzhan Fang.

**Software:** Guangzhan Fang.

**Visualization:** Yanzhu Fan, Shaoxiang Ge.

**Writing – original draft:** Yanzhu Fan, Guangzhan Fang.

**Writing – review & editing:** Yanzhu Fan, Xiaoguo Jiao, Guangzhan Fang.

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
