## [Editor Report · Decision Letter 0]

25 Apr 2025

Dear Dr Fang,

Thank you for submitting your manuscript entitled "The Glutamatergic Projection from the Substantia Nigra Pars Reticulata to the Dorsal Raphe Nucleus Facilitates Social Hierarchy in Mice" for consideration by PLOS Biology.

Your manuscript has now been evaluated by the PLOS Biology editorial staff, and I am writing to let you know that we would like to send your submission out for external peer review. Please note that we would like to consider your article as a Short Report rather than as a Research Article. Please select the Short Report article category when you complete the next step in your submission.

Once your full submission is complete, your paper will undergo a series of checks in preparation for peer review. After your manuscript has passed the checks it will be sent out for review. To provide the metadata for your submission, please Login to Editorial Manager (https://www.editorialmanager.com/pbiology) within two working days, i.e. by Apr 27 2025 11:59PM.

Kind regards,

Taylor

Taylor Hart, PhD,

Associate Editor

PLOS Biology

thart@plos.org

---

## [Decision Letter · Decision Letter 1]

5 Jul 2025

Dear Dr Fang,

Thank you for your patience while your manuscript "The glutamatergic projection from the substantia nigra pars reticulata to the dorsal raphe nucleus facilitates social hierarchy in mice" was peer-reviewed at PLOS Biology. It has now been evaluated by the PLOS Biology editors, an Academic Editor with relevant expertise, and by several independent reviewers.

In light of the reviews, which you will find at the end of this email, we would like to invite you to revise the work to thoroughly address the reviewers' reports.

The reviewers say that the results here are novel and relevant. However, they expressed skepticism on the level of support for the claims that the SNr-DRN pathway has a specific role in social hierarchy, and other areas requiring further controls or analyses. The reviewers have suggested many possible and potentially insightful experiments. However, considering the scope of this Short Report article, we think that it would not be appropriate to require you to perform every suggested experiment. Based on our discussion with the Academic Editor, we think that your revision should focus on providing stronger support for the main findings through new behavioral controls, stronger neuroanatomical characterizations, and addressing the statistical concerns. In particular, you should perform new experiments to address Reviewer 1's point about dissociating exertional effects from social ones, and you should respond thoroughly to all of the points raised by the reviewers.

Given the extent of revision needed, we cannot make a decision about publication until we have seen the revised manuscript and your response to the reviewers' comments. Your revised manuscript is likely to be sent for further evaluation by all or a subset of the reviewers.

We expect to receive your revised manuscript within 3 months. Please email us (plosbiology@plos.org) if you have any questions or concerns, or would like to request an extension. We would be happy to grant you an extension to the revision timeline to allow you to perform more of the experiments suggested by the reviewers.

**IMPORTANT - SUBMITTING YOUR REVISION**

*Re-submission Checklist*

*Published Peer Review*

*PLOS Data Policy*

*Blot and Gel Data Policy*

Sincerely,

Taylor

Taylor Hart, PhD,

Associate Editor

PLOS Biology

thart@plos.org

REVIEWS:

Reviewer #1 [Pegah Kassraian]:

Fan et al. employ the tube test assay, a well-established paradigm, to assess the role of glutamatergic substantia nigra pars reticulata (SNr) neurons and the SNr-to-dorsal raphe nucleus (DRN) pathway in social hierarchy in rodents. This test determines a "winner" in paired trials, where the dominant individual displaces the presumably subordinate counterpart from the confined tube space.

The authors use fiber photometry to characterize the functional relevance of SNr-Glu neurons and apply optogenetic activation and chemogenetic inhibition to dissect the role of the SNr-Glu to DRN pathway. Their findings demonstrate that this pathway bidirectionally modulates social rank, with activation or inhibition respectively promoting or suppressing dominance behavior.

This study is significant as social hierarchy is a fundamental organizing principle across species, yet its neural substrates remain poorly understood. And while for instance the medial prefrontal cortex (mPFC) has been implicated in encoding social rank, the contributions of the SNr and its projections to the DRN have remained largely unexplored.

Thus, results are timely and of great relevance to the field. However, the manuscript requires major revisions, in particular with respect to critical control experiments which are missing and prevent an accurate interpretation of the authors' findings:

Major:

1. The authors conclude that SNr-Glu calcium activity levels are associated with social competition, specifically, rather than general physical effort. However, this claim lacks sufficient experimental support. While the authors demonstrate increased activity during tube test contests, they fail to include critical controls that would dissociate both efforts in a social context and, more specifically, social rank-related efforts from non-social physical exertion. A proper experimental design would require testing SNr-Glu activity during both an active physical effort task without social components (e.g., pushing against an inanimate barrier) and a social but non-competitive effort task. This oversight is particularly significant given the SNr's established role in integrating diverse motor and sensory signals. This fundamental limitation also applies to their optogenetic manipulation experiments of the SNr-Glu to DRN pathway.

2. The study fails to establish how SNr-Glu calcium activity relates to social rank. The fiber photometry data in Fig. 1 only shows event-locked activity during pushing behaviors but provides no information about calcium activity differences between mice of different ranks. This could be furthermore supported by the quantification of the actual physical effort exerted during tube test contests based on more in-depth behavioral analysis, making it possible to determine how fiber photometry calcium activity patterns scale with effort intensity vs rank.

3. The potential lateralization of SNr-Glu signals during social competition remains unexplored, given that the calcium activity signal is related to physical effort, this would be a further important measure to dissociate physical effort from the display of social rank.

4. The interpretation of optogenetic SNr-Glu activation results is equally affected by these limitations regarding effort specificity. While the authors show that stimulation increases tube test success, they do not adequately address whether this reflects a genuine modulation of social hierarchy versus a non-specific increase in physical effort capacity. Moreover, their observation that elevated ranks persist for four days could alternatively be explained by the well-documented "winner effect" in mice, where initial competitive success increases probability of future wins through behavioral reinforcement rather than direct neural encoding of social status, which here could be given rise to by a modulation of physical effectiveness.

5. The authors' conclusion that SNr-Glu activation does not affect anxiety-like behavior, based on elevated plus maze (EPM) results, is potentially compromised by methodological limitations. Their experimental design lacks critical no-light control groups that would account for non-specific effects of light delivery itself, which is known to potentially influence rodent behavior in anxiety assays.

6. The choice to use chemogenetic rather than optogenetic inhibition represents a curious methodological decision that is not adequately justified. Given the distinct temporal dynamics of these techniques (with chemogenetic effects lasting hours compared to milliseconds-scale optogenetic control), the authors' findings of transient rank modulation would benefit from complementary optogenetic inhibition experiments.

7. The apparent discrepancy between cell body and terminal manipulation effects on anxiety-related behavior requires further explanation. While SNr-Glu neuron activation shows no effect on EPM performance, stimulation of their DRN terminals does reduce anxiety-like behavior.

8. The differing temporal profiles of optogenetic versus chemogenetic rank modulation raise important questions about the underlying mechanisms. The authors report that optogenetic effects persist for days while chemogenetic inhibition only transiently alters social status. How do the authors explain this discrepancy?

Minor:

1. Given that the SNr contains predominantly GABAergic neurons and the CaMKIIα promoter is known for leaky expression, it would be good if the authors could provide histological validation that viral expression was largely restricted to glutamatergic neurons (e.g., through co-labeling with VGLUT2 or the like).

2. The exclusive focus on male mice should be explicitly acknowledged in the main text, particularly since the tube test has been successfully implemented in female C57 mice in other studies.

3. Several passages would benefit from clearer language, including: (a) replacing ambiguous terms like "behavioral displays" with more specific descriptions of behaviors, (b) revising "processes" to "neural mechanisms" when referring to hierarchy establishment, (c) changing "expression of dominant behaviors" to "dominance behaviors" for conceptual accuracy, and (d) correcting "aggressive levels" to "aggression levels" throughout for proper grammatical construction. These are just some examples where expression could be improved for clarity.

1. The authors claim that the SNr-Glu "population activity is correlated with active social contests." - however, a critical control - testing if an increase in activity is solely due to the physical effort, or the increase is more specifically tied to physical efforts in social context pertaining to the expression of social rank - are missing. As a test of grip strength is rather an assessment of passive physical effort, the authors need to include a control with an active physical effort only, and, ideally an experimental control where the effort is shown in a social context but not relevant to social rank. This is especially critical, as the SNr acts as locus for the integration of a variety of sensory signals, including those relevant for the execution of motor tasks. Without these controls, the authors conclude that "These findings indicate the relationship between SNr-Glu neuronal activity and social competition and status." does not hold. This criticism also applies to optogenetic manipulations of the SNr-Glu to DRN pathway.

2. Related to point 1, it is not clear how the fiber photometry signal relates to the ranks of the tested individuals. This information is also missing from Figure 1, although it would be important to understand how the mean activity of SNr-Glu neurons differs between mice of different ranks. This analysis might give better insights into concerns raised in point 1, by contrasting the singal evoked related to rank and those evoked by behavioral read-outs of physical efforts, the latter at the time missing from Fig. 1 and from any other analysis in the manuscript. Thus, related to that the authors should quantify measures of "effortful behaviors".

3. Given that the authors link the onset of the physical effort to SNr fiber photometry signal, do they observe any lateralization of the signal?

4. Optogenetic activation of SNR-Glu neurons: This experimental manipulation is subject to the same criticism as articulated under aim 1. Namely, without an appropriate control, it is difficult to interpret results with respect to social rank as they might largely reflect changes in active motor effort. Moreover, while the authors write that "Compared to controls, the significantly elevated social ranks were maintained for at least four days (Fig 2I-2K)". However, it is known that in mice, winning in the tube test increases the propensity of future competitive success ("winner effect"). Thus, in principle, experimental manipulations as conducted here could increase physical effort which induces the winner effect based on the initial success.

5. The authors write "Notably, the duration of time spent in the open arms prior to light activation in the elevated plus-maze (EPM) test was significantly greater than that observed during and after light-on periods; however, no differences in open arm time were found between ChR2-EYFP and EYFP mice (S2A Fig), indicating that optogenetic activation of SNrGlu neurons does not influence anxiety-like behaviors." Given the non-specific effect of light administration, it is not clear how this is to be interpreted. The authors should include no-light control groups into their experimental design.

6. Given the distinct that opto- and chemogenetic manipulations are associated with different timescales where the actuators are active (chemogenetic silencing lasting several hours), it is not clear why the authors have not applied optogenetic inhibition for silencing experiments.

7. The EPM shows no anxiety difference (Supplementary Fig. 2a), but the DRN terminal activation does reduce anxiety (Fig. 4i). This inconsistency suggests SNrGlu cell-body vs. terminal effects may differ.

8. The authors write "The significantly reduced rankings compared to controls were sustained for approximately eight hours; however, by 24 hours post-CNO administration, most mice reverted to their original rank positions (Fig 3I-3K)." How do they explain the difference in modulating ranks across days by using opto- versus cheogenetic methods?

Minor suggestions:

1. Given that some of the cell population in the SNr is also GABAergic, and the CaMKIIa-promoter is known for its leaky expression, have the authors validated that the expression of the GCaMP and other viri is largely in SNr-Glu cells (eg post hoc staining/co-labelling with relevant markers).

2. The tube test has been successfully implemented with female C57 mice. The focus of the authors on male mice only should be stated in the main text of the manuscript.

3. The language is not always clear.

For instance, the authors write in the introduction that "This organizational framework delineates individual rankings within a group, established through recurrent social interactions such as aggressive encounters or behavioral displays". Here, it is not clear what type of behaviors "behavioral displays" refers to.

The authors write "Despite this significance, the processes through which dominance hierarchies are established and maintained remain inadequately understood.", processes is ambigue here - the authors are here not assessing behavioral processes which give rise to dominance hierarchies, but study the underlying neural substrates.

The authors write "and modulating the expression of dominant behaviors in downstream areas" - it should be "dominance behaviors". These are a few examples, but language requires a major revision.

aggressive levels to aggression levels

Reviewer #2: This manuscript entitled "The glutamatergic projection from the substantia nigra pars reticulate to the dorsal raphe nucleus facilitates social hierarchy in mice" investigates the potential involvement of glutamatergic neurons in the substantia nigra pars reticulate (SNr) projecting to the dorsal raphe nucleus (DRN) in regulating social dominance behavior, as assessed by the dominance tube test. Using fiber photometry, the authors demonstrate that SNr glutamatergic (Glu+) neurons are activated during push behavior, but not during mere traversal of the tube without an opponent. Optogenetic activation of SNr Glu+ neurons increased the frequency of push behavior and reduced retreat behavior, as well as enhanced locomotion in the open field test. Conversely, chemogenetic inhibition of SNr Glu+ neurons reduced push behavior, increased retreat, and reduced time spent in the warm spot in a competitive warm spot test. Lastly, the authors show that optogenetic activation of SNr-DRN Glu+ projections is sufficient to increase push behavior in the tube test. This study suggests a novel function of SNr-DRN pathway in effortful push behavior during social competition. However, additional experiments are necessary to support the conclusion that this pathway specifically facilitates social hierarchy.

1) The authors showed a significant reduction in locomotor activity following DREADD-mediated inhibition of SNr Glu+ neurons. Given that the SNr is a well-known brain area involved in motor control (i.e. Partanen & Achim 2022, Front Neurosci), it is possible that the observed reduction in push behavior in the tube test or warm spot test is attributable to motor incoordination rather than a specific deficit in social dominance. To address this concern, please include additional behavioral assays assessing motor coordination (e.g. grip test, rotarod test) to rule out general motor deficits.

2) The relationship between increased push behavior following SNr activation and social dominance requires further clarification. It is possible that SNr Glu+ neurons are generally engaged during any form of effortful motor behavior, not necessarily behaviors associated with social competition. To address this, I recommend recording neural activity during a non-social, effortful task (e.g. pushing a non-animate object) using fiber photometry to determine whether activation of this area is specific to social dominance contexts.

3) The finding that optogenetic activation of the SNr-DRN Glu+ projection enhances push behavior in the dominant tube test without affecting locomotor activity or forelimb muscle strength is promising. However, it remains unclear whether this projection is naturally involved in social dominance behavior. To support this, the authors should perform either optogenetic inhibition of the SNr-DRN projection during the dominance test or fiber photometry recording of this projection during dominance tube test. These experiments would clarify whether endogeneous activity in this pathway is related to dominance behavior.

4) Figure 2 and 4. The authors state "n=24 for light off and n=48 for light on" (Fig. 2D-H) and "n=21 for light off and n=42 for light on" (Fig. 4C-E), but it is unclear how many biological replicates were used in these experiments. Please clarify the number of animals included in each condition. Additionally, it is unclear what exactly is the meant by "light off" and "light on", and why the sample sizes differ between these conditions. If these data represent a within subject comparison, the Wilcoxon signed-rank test should be used instead of Mann Whitney U test. Also, use the average value per animal if multiple data points were obtained from a single animal.

Reviewer #3 [Cliff Kentros]:

Fan and colleagues present a well-written short report investigating the possibility that the relatively understudied glutaminergic neurons of the relatively understudied SNr (as opposed to SNc) may play a specific role in social hierarchy. They support this claim with fiber photometry of the SNr and manipulation of SNr neuronal activity -both via AAVs with the CamKIIa minimal promoter (CK2)- during the tube test, a test of social dominance. There is a lot to like about this study, not least that it is well-written. I do agree that the glutamatergic neurons of the SNr, and indeed the SNr itself, is understudied, and that finding a specific role for them in social hierarchy would be a novel and impactful result, as my understanding is the little that is known specifically about SNr has nothing to do with social interactions. It is an interesting question that to my knowledge nobody has looked at, and they have investigated it with both (admittedly low resolution) physiology and behavior levels. They definitely see behavioral effects of their manipulations, which they are able to modulate up and down, which is always nice. There is something there for sure, and it deserves to be investigated and reported properly.

However, extraordinary claims need more data. My enthusiasm for the paper is limited precisely by the question of specificity, and it pertains to multiple levels. Starting at the behavioral level, the authors are making the claim that their manipulations affect social hierarchy specifically, but almost the only behavioral task that they perform is the tube test, where you put 2 animals at different ends of a tube, and the one that doesn't back out is dominant. The behavioral data in this task seems to be clear, my issue is if all you look at is tasks that measure social hierarchy, then that is the only result that you will get. I do not believe that their discussion of ancillary features of the tube test (e.g. stillness), precludes alternative interpretations of their behavioral results. They should candidly discuss other potential interpretations (anxiety, aggression) of their behavioral results and perform experiments that investigate them. For instance, given that they speak of anxiety effects (which are NOT the same as social effects, though they certainly interact), they need to see if for instance their manipulations are anxiogenic/lytic on other tasks, at the very least the elevated plus maze, which has no social aspects. To be able to say what something is, one needs to show what it is not.

I also find the anatomy kind of sloppy (i.e. not specific enough). The premise of the paper is that SNr glutaminergic neurons exert their effect via dorsal raphe, and while they are indeed connected, it is also true the DRN receives many other inputs (in the atlases they cite SNr is one of many inputs) and that the SNr projects many other places. They use DRN and SNr almost interchangeably throughout the paper, but nearly all measurements and manipulations are in the SNr. The connection to DRN should be downplayed, the current set of results does not sufficiently demonstrate this: the optogenetic stimulation of those fibers locally to the DRN are the only experiments which allow this claim, the chemogenetic ones could be acting elsewhere. So far as I can tell, the basis for the claim of specific manipulations of glutamatergic SNr neurons are the viral promoter, the minimal CK2 promoter. As can even be seen from the panel they show, the overlap between reporter expression with this promoter in AAVs and CK+ neurons is not 100%, and depends upon many things like titre and the precise location of the injections, each of which being arguably unique. The authors need to count the overlap between the expression of their constructs and the SNr CK+ neurons for each animal, and also to know how much expression they got in CK2+ neurons in adjoining regions… that is a crowded neighborhood. Uploading images of each animal would be helpful in this regard, that is what supplementary data is for. I am not expecting overlap to be 100%, and their results would remain interesting even if mixed, but what was manipulated needs to be a known quantity of neurons.

The authors in sum present intriguing preliminary results that with further experimental investigation should become a quite impactful study. The authors seem to have found an interesting behavioral phenotype, though how specifically social it is remains to be determined. It would be a shame if they did not explore it more fully.

---

## [Decision Letter · Decision Letter 2]

29 Jan 2026

Dear Dr Fang,

Thank you for your patience while we considered your revised manuscript "The glutamatergic projection from the substantia nigra pars reticulata to the dorsal raphe nucleus facilitates social hierarchy in mice" for publication as a Short Reports at PLOS Biology. This revised version of your manuscript has been evaluated by the PLOS Biology editors, the Academic Editor, and two of the original reviewers.

Based on the reviews, we are likely to accept this manuscript for publication, provided you satisfactorily address the remaining points raised by Reviewer 3. Please also make sure to address the following data and other policy-related requests.

IMPORTANT: Please ensure that you address all of the following editorial points (bulletted):

**Title:

We would like to tweak your title to better reflect our stylistic preferences. Is the following alternative acceptable to you?

"Glutamatergic projections from the substantia nigra pars reticulata to the dorsal raphe nucleus facilitate male social hierarchies"

We also wondered if the word "regulate" would be preferable to "facilitate". So, a second alternative version of the title would be:

"Glutamatergic projections from the substantia nigra pars reticulata to the dorsal raphe nucleus regulate male social hierarchies"

**Ethics:

-- The Ethics statement needs to be a separate, independent (and the first) subheading in the Material & Methods section. Please also include the specific national or international regulations/guidelines that your animal care and use protocol adhered to. Please note that institutional or accreditation organization guidelines (such as AAALAC) do not meet this requirement. https://journals.plos.org/plosbiology/s/ethical-publishing-practice

**Data:

-- Thank you for making the underlying data available on Dryad. However, it is currently not intuitive to identify the specific file that contains the data shown in a particular plot. To increase accessibility of the data, please do one of the following: a) modify the filenames of the data files to indicate which specific figure panels they are associated with; b) add this information to the README file. This applies to the data underlying the figure panels, at a minimum:

2DEFGHJKL

3CDEFGLMNPQ

4CDEHIJ

S6ABCDE

S7BCDEF

S8ABCD

S9AB

S10ABCDE

S12ABCDE

S13ABCDEFG

-- In addition, S2 Fig appears to contain some extra panels that appear to be duplicates. Please fix.

-- Please also cite the location of the data clearly in all relevant main and supplementary Figure legends, e.g. “The data underlying this Figure can be found in S1 Data” or “The data underlying this Figure can be found in https://doi.org/10.5281/zenodo.XXXXX”

**Code availability:

Per journal policy, if you have generated any custom code during the course of this investigation, we require that you make it available without restrictions. Please ensure that the code is sufficiently well documented and reusable, and that your Data Statement in the Editorial Manager submission system accurately describes where your code can be found.

We expect to receive your revised manuscript within two weeks.

*Published Peer Review History*

*Press*

Sincerely,

Taylor

Taylor Hart, PhD,

Associate Editor

thart@plos.org

PLOS Biology

Reviewer remarks:

Reviewer #2 [Aki Takahashi]: The additional experiments conducted by the authors have satisfactorily addressed all of my concerns.

Reviewer #3 [Clifford Kentros]: The authors present a much improved document in the resubmission, with the addition of several necessary controls and data. The result is a provocative piece of work that I think nicely shows both a novel and interesting set of results and the limitations of behavioral analysis to parse the ramifications of the ever more specific manipulations made possible by modern molecular genetic techniques. In the end, the conclusion of this paper is basically that the glutaminergic neurons of the pars reticulata, unlike its more numerous GABAergic neurons (let alone compacta) which are primarily involved in motor control, play a role specifically in social hierarchy via their innervation of the DRN. That is a remarkable result, really… I would have expected a much more general change (and indeed, there is definitely some crosstalk with neural systems underlying anxiety, see below), but all in all, the authors have done the appropriate controls and the data supports at least a privileged role for this particular projection in social hierarchy.

I personally think they have done sufficient numbers of controls to uphold their main conclusion. All I would ask the authors to do would be to acknowledge the issues and limitations that are indeed inherent to interpreting the behavioral ramifications of network-level manipulations, lest we fall prey to a new kind of phrenology (i.e. this synapse does this, that synapse does that). We all agree to interpret certain behavioral tasks in certain reasonable ways, but need to keep in mind there is some distance between what we measure and the psychological terms we associate with those measurements. In this case, what is clear is their manipulations definitely change the state of the animal in some way that shows up best in those behavioral tests we agree measure social hierarchy and anxiety, but that is not the same as saying this is a pathway that is specific to those ethological and psychological constructs. I know this seems like splitting hairs, but I think it is important to make it explicit.

How to do this? Well, I would say in two ways. First, although they have climbed back a bit from the prior submission, there is still some pretty strong language throughout the document which might lead the reader to think that social hierarchy is all these neurons do, which I don't believe is the authors' intention.

Below is one, but probably not the only example:

"Collectively, these results underscore the critical role of SNrthe glutamatergic neurons SNr-DRN pathway in regulating social hierarchy among male mice." (abstract)

And here's one that says the same thing but doesn't trigger that reaction:

"Collectively, these findings highlight the potential role of the glutamatergic pathway from the SNr to the DRN in the regulation of social hierarchy in male mice." (Intro)

Second, while there are clear examples of them trying (e.g. Line 437), there needs to be a paragraph that reflects the difficulties in interpretation that arise from jumping from the level of stimulating a particular set of neurons to an ethological phenotype. A frank discussion of the fact that, as in their data, anxiety and social position are intertwined, and there are limitations to the interpretation of these tasks from behavioral correlates alone. This leads into their stated plans to investigate these phenomena at a more neurophysiological level, so it shouldn't be hard to incorporate.

Below is the closest discussion part:

"Consistent with these observations, the current findings indicate that the activation of glutamatergic projections from the SNr to the DRN not only enhances social status in mice but also alleviates anxiety levels. This suggests a mutually reinforcing relationship between low-anxious status and the acquisition of dominance. Collectively, these findings highlight the SNrGlu -DRN pathway as a potential neural circuit for modulating anxiety and social dominance in male mice, thereby providing insight into the intricate relationship between social hierarchy and anxiety levels[13]. This relationship may be mediated through distinct neural circuits or receptor mechanisms, emphasizing the need for further research to elucidate the circuitry and synaptic mechanisms underpinning the intrinsic connections between social hierarchy and anxiety levels in mice."

To me, this paragraph still keeps the social aspect of the manipulation paramount… it is entirely possible that it changes in anxiety (or more correctly whatever changes in the brain we measure in those tests) that is driving the social aspect of these changes, that needs to be explicit, or they need to explicitly examine this and make a convincing counter-argument otherwise. They have definitely found something interesting, though, and I congratulate them on that. The differences they obtained regarding local optogenetic manipulation at DRN versus at the SNr are perhaps the most interesting part, in my opinion, as they hint at the complexity of the roles played by these relatively few neurons.

Finally, while this may seem counterintuitive given the above, I would suggest that they have perhaps toned down the language a bit too much in the end of the discussion… yes, its true, more experiments are needed, but these are interesting results!

---

## [Editor Report · Decision Letter 3]

19 Feb 2026

Dear Dr Fang,

Thank you for the submission of your revised Short Reports "Glutamatergic projections from the substantia nigra pars reticulata to the dorsal raphe nucleus regulate male social hierarchies" for publication in PLOS Biology. On behalf of my colleagues and the Academic Editor, Jozsef Csicsvari, I am pleased to say that we can in principle accept your manuscript for publication, provided you address any remaining formatting and reporting issues. These will be detailed in an email you should receive within 2-3 business days from our colleagues in the journal operations team; no action is required from you until then. Please note that we will not be able to formally accept your manuscript and schedule it for publication until you have completed any requested changes.

PRESS

Sincerely,

Taylor

Taylor Hart, PhD,

Associate Editor

PLOS Biology

thart@plos.org